# Effects of ground robot manipulation on hen floor egg reduction, production performance, stress response, bone quality, and behavior

**Guoming Li** [1,2], **Xue Hui** [3], **Yang Zhao** [4‡]*, **Wei Zhai** [5‡], **Joseph L. Purswell** [6‡], **Zach Porter** [1], **Sabin Poudel** [5], **Linan Jia** [5], **Bo Zhang** [5], **Gary D. Chesser** [1]*

**1** Department of Agricultural and Biological Engineering, Mississippi State University, Starkville, Mississippi State, United States of America, **2** Department of Agricultural and Biosystems Engineering, Iowa State University, Ames, Iowa State, United States of America, **3** College of Energy and Intelligent Engineering, Henan University of Animal Husbandry and Economy, Zhengzhou, Henan Province, China, **4** Department of Animal Science, The University of Tennessee, Knoxville, Tennessee State, United States of America, **5** Department of Poultry Science, Mississippi State University, Starkville, Mississippi State, United States of America, **6** USDA, Agriculture Research Service, Poultry Research Unit, Starkville, Mississippi State, United States of America

☯ These authors contributed equally to this work.
‡ YZ, WZ and JLP also contributed equally to this work.
* yzhao@utk.edu (YZ); dchesser@abe.msstate.edu (GDC)

**Data Availability Statement:** The minimal underlining dataset is created and uploaded as Supporting Information files in the submission system.

## Abstract

Reducing floor eggs in cage-free (CF) housing systems is among primary concerns for egg producers. The objective of this research was to evaluate the effects of ground robot manipulation on reduction of floor eggs. In addition, the effects of ground robot manipulation on production performance, stress response, bone quality, and behavior were also investigated. Two successive flocks of 180 Hy-Line Brown hens at 34 weeks of this age were used. The treatment structure for each flock consisted of six pens with three treatments (without robot running, with one-week robot running, and with two-weeks robot running), resulting in two replicates per treatment per flock and four replicates per treatment with two flocks. Two phases were involved with each flock. Phase 1 (weeks 35–38) mimicked the normal scenario, and phase 2 (weeks 40–43) mimicked a scenario after inadvertent restriction to nest box access. Results indicate that the floor egg reduction rate in the first two weeks of phase 1 was 11.0% without the robot treatment, 18.9% with the one-week robot treatment, and 34.0% with the two-week robot treatment. The effect of robot operation on floor egg production was not significant when the two phases of data were included in the analysis. Other tested parameters were similar among the treatments, including hen-day egg production, feed intake, feed conversion ratio, live body weight, plasma corticosterone concentration, bone breaking force, ash percentage, and time spent in nest boxes. In conclusion, ground robot operation in CF settings may help to reduce floor egg production to a certain degree for a short period right after being introduced. Additionally, robot operation does not seem to negatively affect hen production performance and well-being.

**Funding:** Y.Z. N/A Egg Industry Center https://www.eggindustrycenter.org/ The funders (Egg Industry Center) had no role in study design, data collection and analysis, decision to publish, or preparation of the manuscript.

**Competing interests:** The authors have declared that no competing interests exist.

## Introduction

Increased public concern for animal welfare has prompted a growing trend among food retailers to pledge future sourcing of cage-free (CF) eggs exclusively. To meet market demands, the US egg industry is transitioning from conventional cage housing to CF housing [1]. Current CF eggs represent 26% of US egg production [2]. While CF hens are provided with increased living spaces and resources, such as nest boxes, perches, and litter floors [3], eggs laid on the littered floor are a common occurrence in CF systems and negatively impact production performance. Floor egg rates are 0.2–2% of daily egg production and can be over 5% in extreme cases (e.g., lack of nesting training and inadvertent restriction to nest box access) [4]. If not collected in a timely manner, floor eggs are subject to increased risk of bacterial contamination by litter or manure and being eaten or broken by other birds. Delayed collection may also result in habitual floor laying [5]. Pragmatic solutions are needed to reduce floor eggs and prevent economic losses of egg producers.

Several management strategies have been investigated to reduce floor eggs. Young pullets have been supplied with accessible perches [6] and reared with experienced hens [4], so that they are trained to use nest boxes. Reasonable restriction of litter access has also been investigated [4], but may have negative welfare consequences. Enhancing attraction of nest boxes by cleaning and drying nest boxes, providing comfortable substrates inside nest boxes, and creating dim interior environments may be helpful, as well [7, 8].

Robotic applications have increasingly drawn attention in different fields [9]. Some robot manufacturers have focused on designing poultry robot applications towards mitigation of floor eggs in CF housing systems. However, floor egg reduction via robot manipulation has not yet been scientifically and statistically verified. Researchers have investigated robotic application in poultry production and welfare management practices. Yang et al. [10] examined the effects of ground robots on broiler activity, footpad dermatitis, and bird usage of elevated platforms. Parajuli et al. [11] explored broiler and laying hen responses to a ground robot operated at different speeds and frequencies and concluded that the robot did not induce more stress than a human in broiler houses. Dennis et al. [12] tested a mobile ground robot manipulated in a commercial broiler house and reported no influence on broiler behavior expression due to robot manipulation, while mortality and production performance were maintained at an acceptable level. These investigations have demonstrated critical values of robot applications to guide automation and precision management practices in poultry production, but few focused on using robotics as a management tool to reduce floor eggs in CF housing systems.

Several factors and measures should be considered to appropriately evaluate robot operation effects. Preliminary trials were conducted in experimental pens for comprehensively measuring the robot effects before large-scale experiments on commercial farms. This study was primarily interested in the performance of floor egg laying and production indicators under robot treatments. Common physiological stress indicators (i.e., serum corticosterone concentration [13]) should be measured to examine the potential robot-induced stress for laying hens, which was not reported by Parajuli et al. [11]. Yang et al. [10] elaborated that robot operations encouraged broiler movement, and increasing bird activities have been correlated to improved bird bone quality in loose housing systems [3, 14]. Therefore, the robot was assumed to increase hen activities in this study and consequently enhance bird bone quality. Litter moisture may be reduced due to decreased hen lying time and improved ventilation in the aerated litter enhanced by mechanical disturbance during robot turning operations. While not commonly measured in a short-term trial for laying hens, litter moisture content differed significantly from various treatments for 34-week laying hens in an eight-week experiment [15]. Hen footpad health may benefit from the potentially improved litter conditions and can be

reflected by footpad scores determined by footpad dermatitis rates [16]. Although young hens are not likely to have footpad dermatitis, Campbell et al. [17] reported a 0.3% footpad dermatitis rate for free-range hens at 20–36 weeks of age. Ground robots can emit sounds and lights having visual and/or audible alert capabilities and encouraging birds to utilize nest boxes, and nesting behaviors have been used to evaluate the utilization status of nest boxes [18]. With the assistance of the precision agriculture tool, radio frequency identification (RFID) system [19], the interaction of hen nesting behaviors and robot operations can be detected, such as nest box utilization frequency and number of birds simultaneously using a nest box during robot operation.

The objective of this study was to evaluate the effects of robot operation on hen floor egg reduction, production performance, stress response, bone quality, and nesting behavior. Two robot operation durations (one week and two weeks) were investigated to study robot running duration effects on floor egg reduction. Nest boxes were temporarily blocked during the experiment to simulate inadvertent events of nest box restriction in commercial farms.

## Methods

### Animals, housing, and management

This study was conducted at the US Department of Agriculture (USDA) Poultry Research Unit located in Mississippi State. The experimental settings are shown in Fig 1. Two successive flocks of 192 Hy-Line Brown hens at 34 weeks of age were sourced from two commercial farms with aviary housing systems and randomly distributed into six pens. The two flocks were transported in July 2019 and October 2019, respectively. Older birds (34 weeks of age) rather than young pullets were recommended by the industry farm managers, because they may be more likely to overcome the challenges of transportation stress, cold stress (at the end of October), and adaptation to new housing environments. Extra birds were removed after

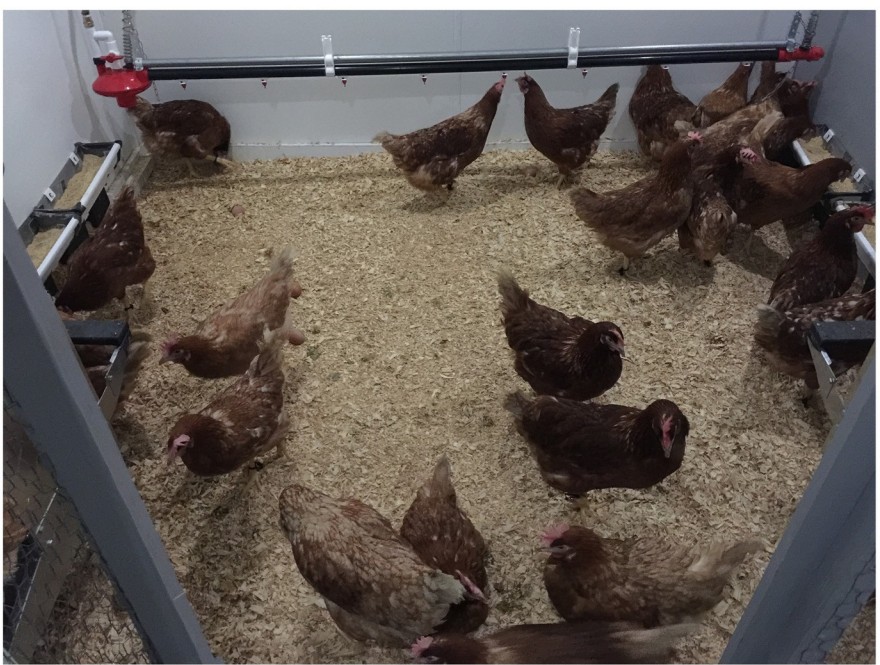 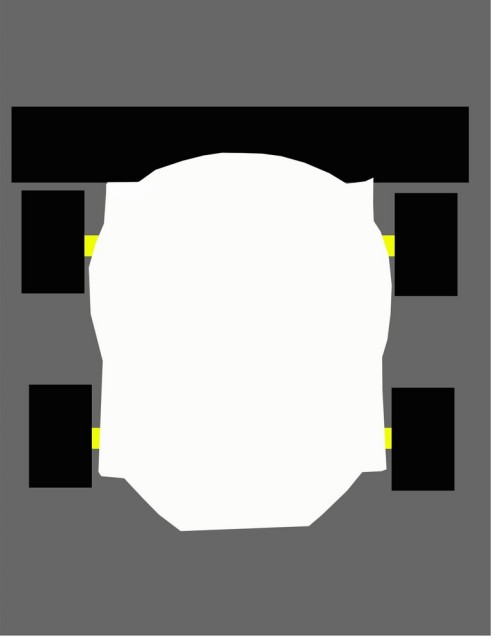

**Fig 1. Illustration of the experimental settings.** A photo of the experimental pen (left) and a schematic drawing of the robot (right).

one-week acclimation, and bird number was equalized to 30 per pen. The six pens were located in an environmentally-controlled space and separated with wooden walls. Each pen was 2.5 m long by 2.2 m wide and equipped with two identical nest boxes (60 cm long, 53 cm wide, and 53 cm high), two feeder troughs, and a suspended nipple drinker line. The nest box space allowance was 212 $cm^2$/hen larger than the 86 $cm^2$/hen in commercial aviary systems and 62 $cm^2$/hen in enriched colony systems [20], and thus should be sufficiently sized for hens in this study. Each nest box contained a red curtain at the entry, a wooden perch in front of the entry, a plastic pad inside, and an egg holder at the back. Nest boxes were affixed to wooden structures at the height of ~25 cm above the floor and installed at the corners of each pen. A night-vision network camera (PRO-1080MSB, Swann Communications USA Inc., Santa Fe Springs, LA, USA) was mounted in the middle of each pen and at ~2 m above the ground. To reduce human interference and prevent adverse events (e.g., severe bird stress caused by the robot) while the robot actively roamed the pen, technicians viewed bird activity from the camera in an adjacent space. Fresh litter at 4-cm depth was spread evenly on the floor before bird arrival. Commercial feed was provided at 7:00 daily and supplemented at 13:00 if feeder troughs were empty. When the animal caretakers performed daily tasks (e.g., feeding birds), they observed bird situations in each pen and recorded unusual events (e.g., bird pecking). Temperature, lighting program, and light intensity were, respectively, set to 24˚C, 16L:8D (light ON at 6:00 and OFF at 22:00), and 20 lux at bird level. The animal use of this study was conducted according to the guidelines of the USDA-ARS Animal Care and Use Committee at Mississippi State (protocol 19–6 and date of approval 27 June 2019).

## Experimental design and robot treatments

The six pens were randomly assigned with three treatments (Fig 2), with two replications per treatment in each flock resulting in four replications per treatment across the two flocks. The three treatments included 1) without robot running, 2) with one-week robot running, and 3) with two-week robot running. For each flock, the experiment lasted ten weeks and consisted of two phases. Phase 1 was from weeks 35 to 39. In the robot treatment pens, a robot (50×60×15 cm, L×W×H) was operated from Monday to Friday of each week. On robot-operation days, the robot automatically roamed the entire floor for 5 min every half hour from 7:00

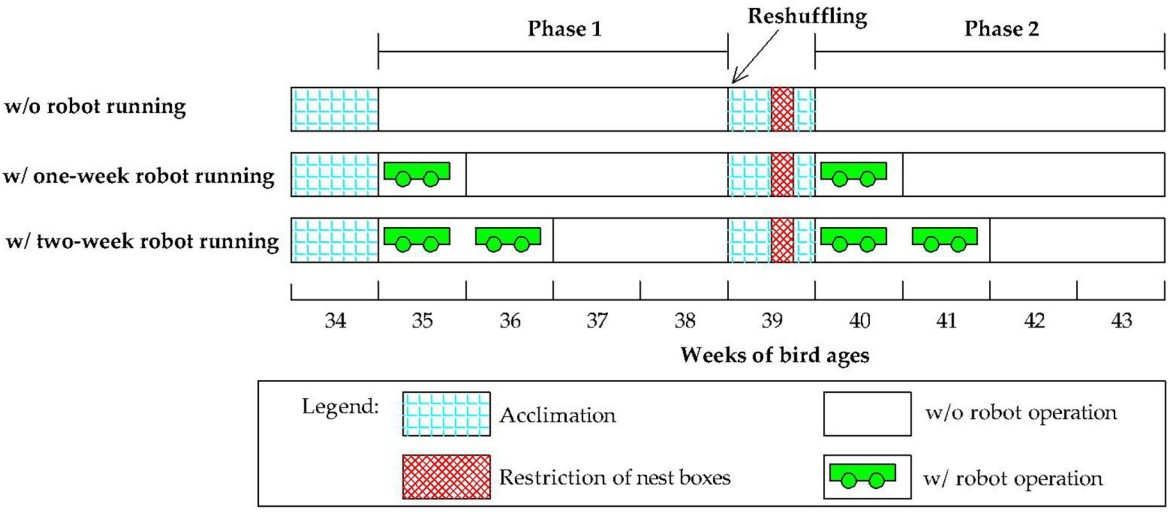

**Fig 2. Schematic drawing of experimental design.** "w/o" and "w/" are without and with, respectively.

to 13:00, which covered the major egg-laying period in a day, resulting in a total of 350 run min in a week. The robot roamed at a 0.16 m/s linear velocity to minimize robot-induced hen stress [11] and changed routes automatically once touching a bird in its path. Ten birds in each pen fitted with two RFID tags (Section 2.3.4 Nesting behavior monitoring) on one leg were used for blood and bone sampling.

Phase 2 (40–43 weeks of bird age) was designed to investigate the floor egg reduction via robot after nest box restriction. The hens in phase 1 were also used in phase 2. Before phase 2, birds in the six pens were reshuffled to reduce the influence of the treatments in phase 1. The ten tagged birds were kept in the original pens while the other 20 birds were equally reassigned to the rest of the pens. After the reshuffle, hens were acclimated to the environment for one week, with two days of nest box access restriction. The treatments were implemented in the six pens in the subsequent four weeks following the same procedures in phase 1.

## Data collection

**Production, environment, and footpad conditions.** Floor and nest eggs were manually collected daily throughout the two flocks, while they were weighed separately every day from weeks 39 to 43. However, due to unexpected events, egg mass data in flock 1 was lost, and only flock 2 data were used for further analysis. Feed in feed bins and feeder troughs of each pen were weighed weekly. At the end of each flock, live body weights were collected for each pen, and three litter samples were randomly collected from the littered floor in each pen to obtain litter moisture content. Each sample (50–90 g) was collected from the surface layer at 0–4 cm depth and weighed immediately after collection and again after oven-drying at 105°C for 24 hours using an analytical balance (0.01 g precision), in accordance with ASAE Standard 358.3 [21]. Footpad health of each bird was evaluated at the end of each flock based on the Welfare Quality Assessment Protocol for Poultry [16] by two trained observers, and evaluation results were verified between observers. Dead birds in each pen were recorded and removed during daily inspection.

**Blood sampling and analysis.** Blood sampling was conducted at the end of weeks 34 and 38. Week 34 corticosterone concentrations were considered baselines after birds adapted to the new housing. Six out of ten tagged birds were randomly selected from each pen and bled, resulting in a total of 24 sampled birds per treatment. In week 38, the same birds were used for blood withdrawal. All birds for blood sampling were handled similarly to avoid introducing additional stress during collection. The physiological parameters (i.e., heart rate, respiratory rate, body temperature, and mucous membrane color) were observed and detected by the poultry scientists before blood sampling. This approach ensured that all assessed birds were healthy and minimized confounding factors or pathological conditions that would otherwise affect the results. The blood sample of each assessed bird was collected from the brachial vein and placed in a 10 mL BD Vacutainer tube without anticoagulant. A sample was usually less than 1% of body weight to maintain bird health after collection. Blood samples were set in a water bath (at 37°C) for 2 hours to clot and then centrifuged at 3,500 rpm and 4°C using a refrigerated centrifuge (model J-6B, Beckman Coulter, Inc., Brea, CA, USA) for 20 minutes to extract serum. Serum samples were stored in a -80°C freezer for further analysis. The Chicken Corticosterone Enzyme-Linked Immunoassay Kit (MBS282652, MyBioSource company, San Diego, CA, USA) was used for the serum samples. The plate absorbance was measured at a wavelength of 450 nm (BioTek Cytation1 Imaging reader). Absorbance results were analyzed to determine the concentration of corticosterone present in the serum sample using a four-parameter logistic curve [22]. Based on the detection range of the kit, the calculated concentrations of <50 pg/ml or >10000 pg/ml were treated as outliers and ruled out for the analysis.

**Bone sampling and testing.**    At the end of the experiments, birds were humanly eutha-
nized according to the AVMA Guidelines [23], and tibia bone samples of the ten RFID tagged
birds in each pen were collected. General measures of bone quality include bone breaking
force and bone ash content [24]. Bone breaking force was determined using a Universal Test-
ing Instrument (Instron 5544, Instron ltd, Norwood, MA, USA). Ash contents of the tibia
bones were determined after the breaking force test. Each broken bone was weighed with the
same analytical balance and oven-dried for 24 hours at 120˚C. The oven-dried bone was
weighed, placed in a crucible, and oven-dried again in a furnace (Isotemp D3714 muffle fur-
nace, Thermo Fisher Scientific Inc., Waltham, MA) for 24 hours at 600˚C. After the last dry-
ing, the remaining content was weighed to obtain ash weights of bones. All procedures were
followed as described by Standard S459 of the American Society of Agricultural and Biological
Engineers (ASABE) [25].

**Nesting behavior monitoring.**    Bird nesting behaviors were registered by a ultra-high fre-
quency RFID system [19]. The antennas (TIMES-7 A6034S, Impinj Inc., Seattle, WA, USA)
connected with the reader (IPJ-REV-420, TransTech Systems Inc., Wilsonville, OR, USA) via
ethernet cables were affixed to the top of the nest boxes, and passive RFID tags (PT-103, Trans-
Tech Systems Inc., Wilsonville, OR, USA) were attached to the legs of birds using zip ties.
After the hardware was installed, an RFID tag was held near the antenna, and the interface of a
free visualization software, MultiReader for SpeedWay Gen2 RFID (Version 6.6.11.240), was
observed. Once the RFID tag was within the detection range of the antenna, readings were dis-
played on the interface and the system power was adjusted, so that the system was validated to
only record birds located inside a nest box rather than birds standing next to a nest box. The
free software was only for real-time data visualization and could not store validation data. It
should be noted that passive tags neither have an internal power source nor actively emit sig-
nals. They utilize electromagnetic waves received from a reader. Once the reader transmits sig-
nals to a tag, the connected antenna creates a magnetic field and the tag circuit uses the power
generated to transmit data back to the reader. Theoretically, a passive tag does not have a
detection range, but the connected antenna had one within 80 cm based on current RFID set-
tings and power supply. Birds in the nest boxes were continuously monitored throughout
every second of the experiments and stored in a Python-based data acquisition system. Bird
ID, presence time, and nest box ID were consistently saved into CSV files for further analysis.

## Parameter calculation and definition

**Floor eggs.**    Weekly floor egg rates (%) from weeks 34 to 43 were calculated in Eq 1 and
reflected the change in floor eggs over time. In week 39, only the floor eggs after nest
box reopening were accounted for to calculate the weekly floor egg rate. In Eq 2, relative floor
egg reduction was obtained by comparing the floor egg rates between weeks 35–38 (tested
ones) and week 34 (initial one) and between weeks 40–43 (tested ones) and week 39 (initial
one). The relative floor egg reduction was to normalize the reduction and reduce the effects of
various initial floor egg rates in each pen.

$$\text{Weekly floor egg rate }[\%] = 100 \times \frac{\textit{Number of floor eggs in a week}}{\textit{Number of eggs in a week}} \tag{1}$$

$$\text{Relative floor egg reduction }[\%] = 100 \times \frac{\textit{Intial weekly floor egg rate} - \textit{Tested weekly floor egg rate}}{\textit{Intial weekly floor egg rate}} \tag{2}$$

**Production performance, footpad health condition, and litter condition.** The number of birds in a pen was initially corrected based on the number of dead birds and corresponding duration (Eq 3). Hen-day egg production (%), feed intake (g/bird/day), and feed conversion ratio (FCR, kg feed/dozen eggs) were calculated from weeks 34 to 43 (Eqs 4–6). An alternative metric exists for FCR, kg feed/kg eggs, but we only had egg mass in weeks 39–43 of flock 2 and thus the alternative metric was not considered in this case. Egg mass of nest eggs, floor eggs, and the sum of both was obtained in weeks 40–43. Live body weight (kg/bird) was obtained by dividing group weight with the number of birds in a pen (Eq 7), and litter moisture content (%) was calculated by comparing fresh litter weight and dried litter weight (Eq 8). These two parameters were only obtained for week 43. Footpad score and mortality were evaluated based on the assessment protocol described earlier.

$$N_{avg} = N_0 - \frac{\sum_{i=1}^{T} M_i \times (T - i + 1)}{T} \tag{3}$$

$$\text{Hen} - \text{day egg production } [\%] = \frac{Number\ of\ eggs\ in\ a\ week}{7 \times N_{avg}} \times 100\% \tag{4}$$

$$\text{Feed intake } [g/bird/day] = \frac{Feed\ consumption\ in\ a\ week}{7 \times N_{avg}} \tag{5}$$

$$\text{Feed conversion ratio } [kg\ feed/dozen\ eggs] = \frac{Feed\ consumption\ in\ a\ week}{\frac{1}{12} \times Number\ of\ eggs\ in\ a\ week} \tag{6}$$

$$\text{Live body weight } [kg/bird] = \frac{Group\ weight}{N_{avg}} \tag{7}$$

$$\text{Litter moisture content } [\%] = \frac{Weight\ of\ fresh\ litter - Weight\ of\ dried\ litter}{Weight\ of\ fresh\ litter} \times 100\% \tag{8}$$

where $N_{avg}$ is time weighted bird number; $N_0$ is initial bird number (30 in this case); $T$ is maximum duration of period in days; $M_i$ is number of mortalities on the $i^{th}$ day; and $i$ is day of period. Litter moisture content is wet basis moisture content.

**Stress response and bone quality.** Serum corticosterone concentrations (pg/ml) were used to reflect bird stress response to the robot, and the mean concentrations were obtained by averaging the concentrations of individual birds in a pen. Bone quality was indicated by bone breaking force (kg), fresh bone weight (g), dried bone weight (g), bone ash weight (g), and ash percentage (%) (Eq 9). Mean values of these parameters were obtained from individually measured birds in a pen. The coefficient of variations (CVs) of these parameters was calculated by dividing standard deviations of the parameters with the mean values (Eq 10).

$$\text{Ash percentage } [\%] = \frac{Ash\ weight}{Dried\ weight} \times 100\% \tag{9}$$

$$\text{Coefficient of variations } [\%] = \frac{Standard\ deviation}{Mean} \times 100\% \tag{10}$$

**Nesting behavior.** The collected RFID data were processed to extract nesting behaviors. Nesting behaviors were considered only in the photoperiod (6:00 to 22:00). Time spent in nest boxes was summarized based on individual birds from 7:00 to 13:00 (min/bird) and in a day (min/bird/day). Frequency (%) of number of birds simultaneously using a nest box was also obtained for the two durations. Hourly time spent in nest boxes (min/bird/h) was obtained by summing the time across each hour of the photoperiod.

## Statistics

The experimental unit was the pen, and the effects of the robot treatment, bird age, and interaction of both were repeatedly measured with pens. Egg mass was measured from weeks 40 to 43 in flock 2. The robot treatment effects on all parameters were analyzed with the ANOVA using the PROC MIXED statement in Statistical Analysis Software (SAS 9.3, SAS Institute Inc., Cary, NC., USA). Treatment means and CV were compared using Fisher's least significant difference multiple post-hoc comparisons, with significance considered at P≤0.05. A constant of 1 was added to all percentage data in decimal form to eliminate negatives and then log transformed to approach normality. The statistical results (least square mean and standard error) were back transformed into percentages. The sample size (n) for each treatment per pen was 40 for weekly floor egg rate, hen-day egg production, feed intake, FCR, and time spent in nest boxes; 32 for relative floor egg reduction; 24 for nest and floor egg mass; 4 for live body weight, footpad score, and mortality; 6 for litter moisture content; and 8 for serum corticosterone concentrations, bone breaking force, fresh bone weight, dried bone weight, bone ash weight, and ash percentage.

## Results

No interaction effects of robot treatment and bird age were observed across all measures (P≥0.05). Therefore, detailed statistical results about interaction effects are not reported in the following tables and figures.

### Floor egg reduction and floor egg rate

Weekly floor egg rate and relative floor egg reduction are presented in Table 1. The average weekly floor egg rate was 31.6–59.6%, and the average relative floor egg reduction was 21.1–41.8%. Overall, the robot treatment had no effect on these two parameters (P≥0.57), but the weekly floor egg rate decreased from weeks 34 to 38 and from weeks 39 to 43 (P<0.01). During weeks 35–36, the floor eggs were reduced more with two-week robot running than other treatments. The floor eggs were reduced slowly during weeks 37–38 without treatments. In phase 2, the floor eggs in all pens were reduced 7.3–12.2% in week 40 compared to those in week 39, while remaining nearly unchanged in the subsequent weeks.

### Production performance, footpad health condition, and litter condition

**Hen-day egg production, feed intake, and feed conversion ratio.** Hen-day egg production, feed intake, and feed conversion ratio in the two experimental phases are presented in Table 2. The averages of the three parameters were 85.9%, 130.0 g/bird/day, and 1.78 kg feed/dozen eggs, respectively. The three parameters were not significantly different among treatments for both experimental phases (P≥0.67). The three parameters changed with bird ages (P<0.01). Hen-day egg production was lower than 80% in week 34 but gradually increased or slightly fluctuated as time went on. Feed intake greatly fluctuated during weeks 34 to 43, and increments were observed between weeks 36 and 38 and between weeks 39 and 41. FCR

**Table 1. Weekly floor egg rates and relative floor egg reduction.**

| Treatment | Weekly floor egg rate (%) | Relative floor egg reduction (%) |
|---|---|---|
| Robot treatment | | |
| w/o robot running | 32.1 | 35.4 |
| w/ one-week robot running | 46.0 | 27.1 |
| w/ two-week robot running | 40.7 | 35.1 |
| SEM[1] | 6.8 | 12.0 |
| Bird age | | |
| 34 | 59.6[a] | – |
| 35 | 47.3[b] | 21.1 |
| 36 | 38.3[c] | 35.0 |
| 37 | 36.9[c] | 38.7 |
| 38 | 34.7[c] | 41.8 |
| 39 | 53.4[ab] | – |
| 40 | 31.7[c] | 32.1 |
| 41 | 32.0[c] | 30.5 |
| 42 | 32.5[c] | 29.7 |
| 43 | 31.6[c] | 32.3 |
| SEM[2] | 4.4 | 7.6 |
| P-value | | |
| Robot treatment | 0.57 | 0.90 |
| Bird age | <0.01 | 0.12 |
| Robot treatment × Bird age | 0.94 | 0.68 |

**Note:** [a,b,c] Values having different superscripts within the same treatment are significantly different, with a significance level of 0.05, least square difference post-hoc analysis, and PROC MIXED statement. [1] Standard errors are for the effect of robot treatment (n = 40 for weekly floor egg rate and n = 32 for relative floor egg reduction) and [2] Standard errors are for the effect of bird age (n = 12 for both parameters). 'w/' and 'w/o' represent with and without, respectively. "–" indicates missing information. The floor egg reduction in weeks 35–38 and weeks 40–43 was compared with weeks 34 and 39, respectively.

decreased between weeks 34 and 35, increased between weeks 37 and 39, and slightly fluctuated around 1.87 kg feed/dozen eggs in the following weeks.

**Egg mass.** Egg mass is presented in Table 3 and Fig 3. The eggs in weeks 40 to 43 were 60.2–63.5 g. The nest egg mass with two-week robot running was significantly smaller than that with other treatments (P = 0.05), while floor egg mass was not different among treatments (P = 0.8) (Table 3). In addition, a significant reduction in mean floor egg mass relative to mean nest egg mass was observed (P = 0.05) (Fig 3).

**Live body weight and litter moisture content.** Live body weight and litter moisture content in week 43 are presented in Fig 4. Average body weight and litter moisture content were 1.84 kg/bird and 16.5%, respectively. They were also not significantly different among the treatments (P≥0.38).

**Footpad score and mortality.** Zero footpad scores indicate perfect footpad condition for all birds with different treatments (Table 4). Only one or two mortalities were observed during each flock.

## Stress response

Table 5 shows the mean and CV of serum corticosterone concentrations. The mean serum corticosterone concentrations were 138.4–642.9 pg/ml, and CVs ranged from 24.9–97.1%. The

**Table 2. Hen-day egg production, feed intake, and feed conversion ratio (FCR).**

| Treatment | Hen-day egg production (%) | Feed intake (g/bird/day) | FCR (kg feed/dozen eggs) |
|---|---|---|---|
| Robot treatment | | | |
| w/o robot running | 86.0 | 132.4 | 1.78 |
| w/ one-week robot running | 85.4 | 127.3 | 1.75 |
| w/ two-week robot running | 85.3 | 103.4 | 1.80 |
| SEM[1] | 2.1 | 15.4 | 0.08 |
| Bird age | | | |
| 34 | 77.3[c] | 123.2[bc] | 1.76[b] |
| 35 | 86.8[ab] | 108.3[cd] | 1.43[c] |
| 36 | 85.9[ab] | 103.9[d] | 1.46[c] |
| 37 | 80.5[bc] | 125.3[bc] | 1.92[ab] |
| 38 | 89.4[a] | 151.6[a] | 1.86[ab] |
| 39 | 90.9[a] | 103.7[d] | 1.79[ab] |
| 40 | 86.3[ab] | 115.8[bcd] | 1.81[ab] |
| 41 | 86.1[ab] | 127.8[b] | 1.87[ab] |
| 42 | 85.2[ab] | 119.7[bcd] | 1.90[ab] |
| 43 | 88.3[a] | 131.0[b] | 1.95[a] |
| SEM[2] | 1.7 | 10.6 | 0.08 |
| P-value | | | |
| Robot treatment | 0.99 | 0.40 | 0.92 |
| Bird age | <0.01 | <0.01 | <0.01 |
| Robot treatment × Bird age | 1.00 | 0.74 | 1.00 |

**Note:** [a,b,c,d] Values having different superscripts within the same treatment are significantly different, with a significance level of 0.05, least square difference post-hoc analysis, and PROC MIXED statement. [1] Standard errors are for the effect of robot treatment (n = 40) and [2] Standard errors are for the effect of bird age (n = 12). 'w/' and 'w/o' represent with and without, respectively.

means and CVs were not significantly different among treatments in weeks 34 and 38 (P≥0.07), while the mean concentration in week 38 was 232.7 pg/ml higher than that in week 34.

## Bone quality

Table 6 shows the means of parameters for evaluating bone quality of hens. The average bone breaking force, fresh bone weight, dried bone weight, bone ash weight, and ash percentage were 24.8 kg, 11.0 g, 7.3 g, 4.1 g, and 56.0%, respectively, and they were not significantly different among treatments (P≥0.20).

**Table 3. Egg mass under the treatments.**

| Treatment | Nest egg mass (g/egg) | Floor egg mass (g/egg) | Overall (g/egg) |
|---|---|---|---|
| w/o robot running | 62.9[ab] | 60.2 | 62.6[ab] |
| w/ one-week robot running | 63.5[a] | 62.2 | 63.2[a] |
| w/ two-week robot running | 61.7[b] | 62.2 | 61.7[b] |
| SEM | 0.5 | 0.8 | 0.4 |
| P-value | 0.05 | 0.13 | 0.04 |

**Note:** SEM is standard error of the mean (n = 8). Data were from 40–43 weeks of bird ages in flock 2. Values having different superscripts are significantly different, with a significance level of 0.05, least square difference post-hoc analysis, and PROC MIXED statement. 'w/' and 'w/o' represent with and without, respectively.

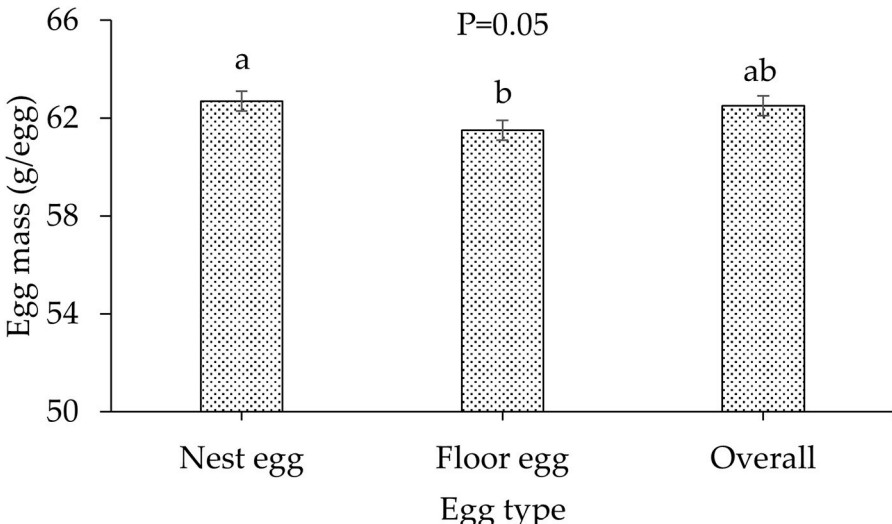

**Fig 3. Illustration of egg mass of different egg types.** Data were from 40–43 weeks of bird ages in flock 2. Values having different letters atop the bars are significantly different (n = 24), with significance level of 0.05, least square difference post-hoc analysis, and PROC MIXED statement.

Table 7 shows the CVs of the parameters in week 43 for evaluating bone quality. The average CVs were 26.4% for bone breaking force, 9.5% for fresh bone weight, 11.3% for dried bone weight, 13.5% for bone ash weight, and 5.5% for ash percentage, and they were not significantly different among treatments (P≥0.31).

## Nesting behaviors

**Time spent in nest boxes.** Table 8 presents time spent in nest boxes during the daily robot running period and across the day by treatments. Overall, hens spent 14.1 min in nest boxes daily, out of which nearly 50% was from 7:00 to 13:00. The birds under robot treatments did not spend significantly different amounts of time in nest boxes during daily robot running periods and across the day (P≥0.52). Birds changed their nesting time significantly among bird ages (P<0.01). The hens with two-week robot running spent more time in nest boxes during weeks 35–37 than those with other treatments, while the time spent was similar in other

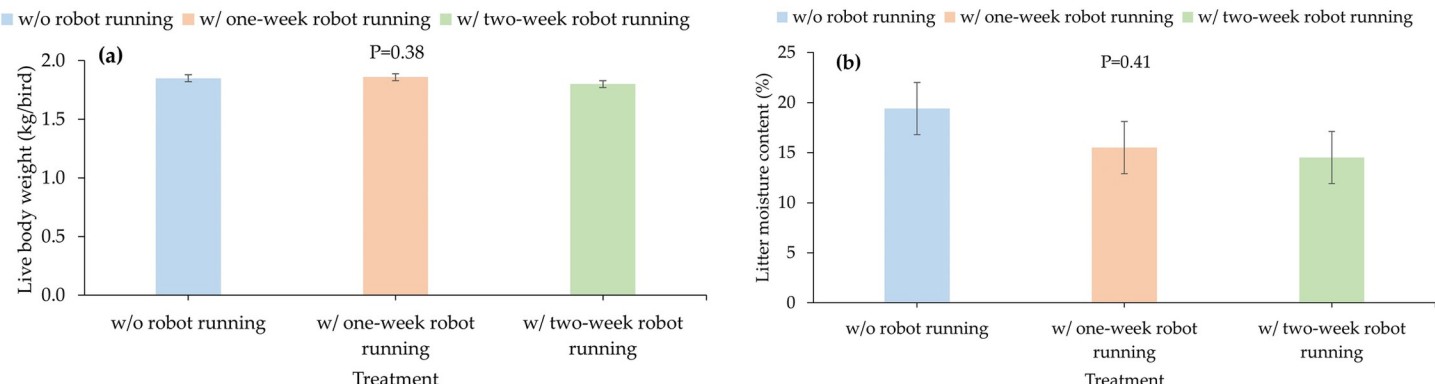

**Fig 4.** Live body weight (a) (n = 4) and litter moisture content (b) (n = 6) under the treatments. Data were only measured at the end of the experiments (43 weeks of bird age). 'w/' and 'w/o' represent with and without, respectively.

**Table 4. Footpad score and mortality.**

| Treatment | Footpad score | Mortality (%) |
|---|---|---|
| w/o robot running | 0 | 0.8 |
| w/ one-week robot running | 0 | 0.0 |
| w/ two-week robot running | 0 | 1.6 |
| SEM | – | 0.7 |
| P-value | – | 0.32 |

**Note:** SEM is standard error of the mean (n = 4). Data were only obtained at the end of the experiments. 'w/' and 'w/o' represent with and without, respectively. "–" indicates missing value.

**Table 5. Mean and coefficient of variations (CV) of serum corticosterone concentrations.**

| Treatment | Mean (pg/ml) | CV (%) |
|---|---|---|
| Robot treatment | | |
| w/o robot running | 251.5 | 55.4 |
| w/ one-week robot running | 400.8 | 80.7 |
| w/ two-week robot running | 146.8 | 42.1 |
| SEM[1] | 93.4 | 6.6 |
| Bird age | | |
| 34 | 150.0 | 66.4 |
| 38 | 382.7 | 51.2 |
| SEM[2] | 77.7 | 5.0 |
| P-value | | |
| Robot treatment | 0.21 | 0.07 |
| Bird age | 0.07 | 0.17 |
| Robot treatment × Bird age | 0.28 | 0.06 |

**Note:** [1] Standard errors are for the effect of robot treatment (n = 8) and [2] Standard errors are for the effect of bird age (n = 12). 'w/' and 'w/o' represent with and without, respectively.

**Table 6. Means of bone breaking force, fresh bone weight, dried bone weight, bone ash weight, and ash percentage under the treatments.**

| Treatment | Bone breaking force (kg) | Fresh bone weight (g) | Dried bone weight (g) | Bone ash weight (g) | Ash percentage (%) |
|---|---|---|---|---|---|
| Robot treatment | | | | | |
| w/o robot running | 24.5 | 11.2 | 7.4 | 4.1 | 56.1 |
| w/ one-week robot running | 25.0 | 10.7 | 7.1 | 4.0 | 56.0 |
| w/ two-week robot running | 24.7 | 11.1 | 7.3 | 4.1 | 55.8 |
| SEM[1] | 0.8 | 0.2 | 0.1 | 0.1 | 0.5 |
| Bird age | | | | | |
| 34 | 24.2 | 10.9 | 7.2 | 4.0 | 56.2 |
| 42 | 25.3 | 11.1 | 7.3 | 4.1 | 55.7 |
| SEM[2] | 0.6 | 0.2 | 0.1 | 0.1 | 0.4 |
| P-value | | | | | |
| Robot treatment | 0.89 | 0.20 | 0.27 | 0.64 | 0.98 |
| Bird age | 0.22 | 0.39 | 0.44 | 0.83 | 0.67 |
| Robot treatment × Bird age | 0.78 | 0.23 | 0.32 | 0.66 | 0.96 |

**Note:** [1] Standard errors are for the effect of robot treatment (n = 8) and [2] Standard errors are for the effect of bird age (n = 12). 'w/' and 'w/o' represent with and without, respectively.

**Table 7. Coefficient of variations (CV) of bone breaking force, fresh bone weight, dried bone weight, bone ash weight, and ash percentage under the treatments.**

| Treatment | Bone breaking force (%) | Fresh bone weight (%) | Dried bone weight (%) | Bone ash weight (%) | Ash percentage (%) |
|---|---|---|---|---|---|
| w/o robot running | 24.1 | 9.2 | 11.4 | 13.5 | 5.4 |
| w/ one-week robot running | 27.5 | 9.2 | 11.0 | 13.4 | 5.5 |
| w/ two-week robot running | 27.6 | 9.5 | 11.4 | 13.6 | 5.4 |
| SEM | 1.4 | 0.6 | 0.4 | 0.4 | 0.4 |
| P-value | 0.31 | 0.88 | 0.77 | 0.97 | 0.98 |

**Note:** SEM is standard error of the mean (n = 4). 'w/' and 'w/o' represent with and without, respectively. The data was from week 43.

bird ages. The overall trend with the three treatments showed that hourly time spent in nest boxes increased from weeks 34 to 35, decreased from weeks 36 to 38, and stabilized or slightly changed in the following weeks.

**Frequency of number of birds simultaneously using a nest box.** Fig 5 shows the frequency of number of birds simultaneously using a nest box with regard to time periods (7:00 to 13:00 and a whole day). The pattern of the frequency corresponding to different

**Table 8. Hen time spent in nest boxes under the treatments.**

| Treatment | Time spent in nest boxes from 7:00 to 13:00 (min/bird/day) | Time spent in nest boxes during a day (min/bird/day) |
|---|---|---|
| Robot treatment | | |
| w/o robot running | 6.6 | 12.9 |
| w/ one-week robot running | 6.2 | 13.0 |
| w/ two-week robot running | 6.9 | 16.1 |
| SEM[1] | 0.6 | 2.2 |
| Bird age | | |
| 34 | 7.4[ab] | 14.6[bc] |
| 35 | 6.8[bcd] | 14.9[b] |
| 36 | 8.6[a] | 19.2[a] |
| 37 | 7.0[bc] | 15.4[b] |
| 38 | 6.1[cd] | 14.4[bcd] |
| 39 | 6.1[cd] | 13.0[bcd] |
| 40 | 6.3[bcd] | 13.2[bcd] |
| 41 | 6.0[cd] | 12.5[bcd] |
| 42 | 5.7[d] | 11.3[d] |
| 43 | 5.8[cd] | 11.4[cd] |
| SEM[2] | 0.6 | 1.6 |
| P-value | | |
| Robot treatment | 0.77 | 0.52 |
| Bird age | <0.01 | <0.01 |
| Robot treatment × Bird age | 0.39 | 0.74 |

**Note:** [a,b,c,d] Values having different superscripts within the same treatment are significantly different, with a significance level of 0.05, least square difference post-hoc analysis, and PROC MIXED statement. [1] Standard errors are for the effect of robot treatment (n = 40) and [2] Standard errors are for the effect of bird age (n = 12). 'w/' and 'w/o' represent with and without, respectively.

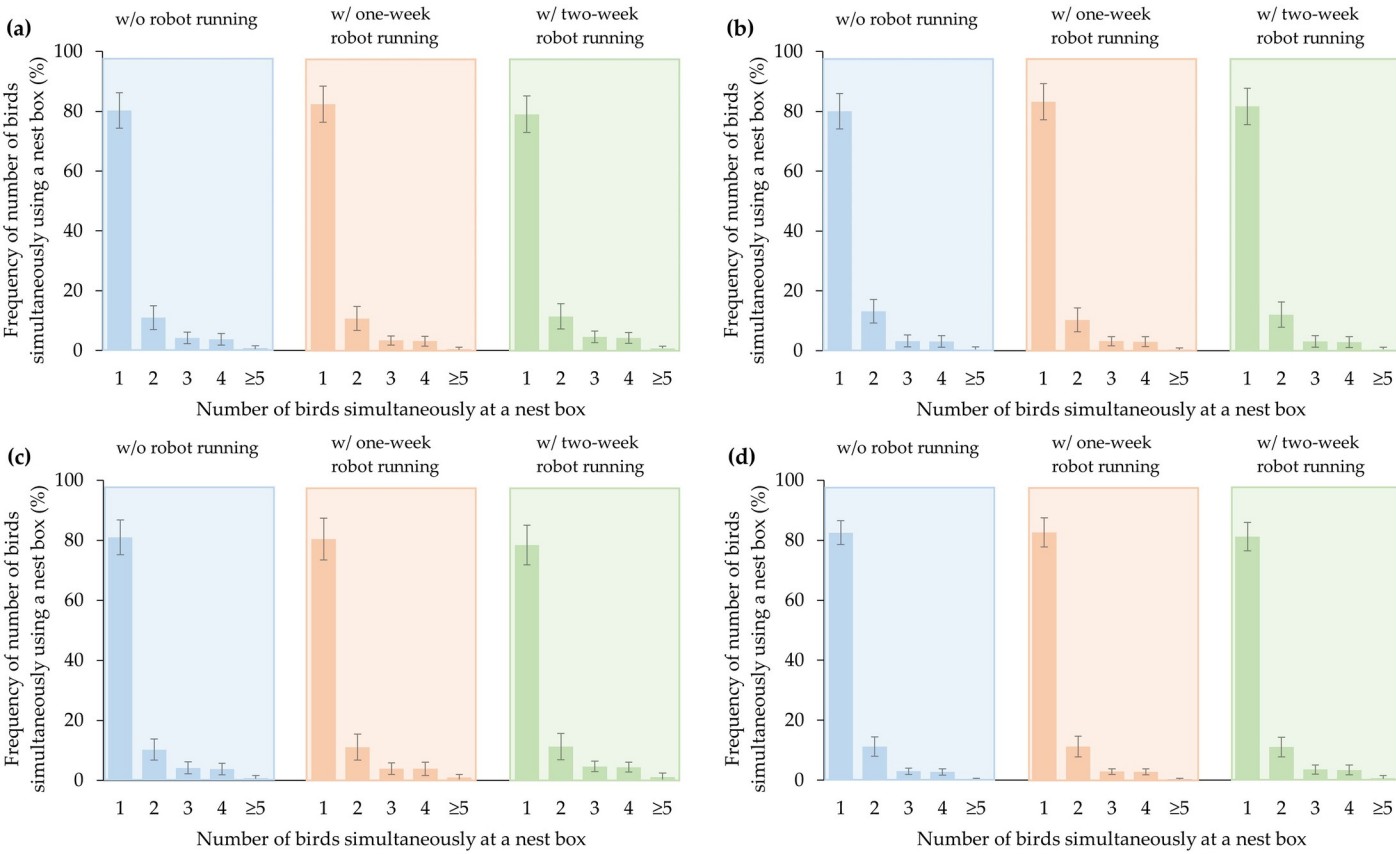

**Fig 5.** Frequency of number of birds simultaneously using a nest box with regard to different sections of time: (a) during 7:00 to 13:00 in weeks 35–38; (b) during 7:00 to 13:00 in weeks 40–43; (c) in a day in weeks 35–38; and (d) in a day in weeks 40–43. 'w/' and 'w/o' represent with and without, respectively.

bird numbers was quite similar for both 7:00 to 13:00 and a whole day, across the two experimental phases. The cases of one bird presenting in a nest box took up nearly 80% of the total cases, while the cases of over five birds simultaneously using a nest box were less than 1%. Notably, the frequency was only counted when at least one bird was using a nest box.

**Hourly time spent in nest boxes.** Hourly time spent in nest boxes is presented in Fig 6. The hens spent 0.3–1.6 min in nest boxes each hour of 6:00–21:00, while they spent 8.6–14.8 min at 21:00. The overall trend with the three treatments showed that hourly time spent in nest boxes increased from 6:00 to 8:00, gradually decreased from 9:00 to 19:00, and sharply increased again from 20:00 to 21:00.

## Anecdotal observations for bird adaptation to the robot

Birds were familiarized with robots as time went on (Fig 7). The hens were startled and clustered together when the robot was first placed in the pen. After three days of robot running, birds behaved normally but kept a distance from the robot. After seven days of robot running, birds often remained in close proximity to the robot. At the end of the two-week robot running period, birds sometimes rode the robot as it roamed around the pen. Video data in flock 1 were lost because of unexpected events, and only anecdotal observations rather than quantitative statistics were presented.

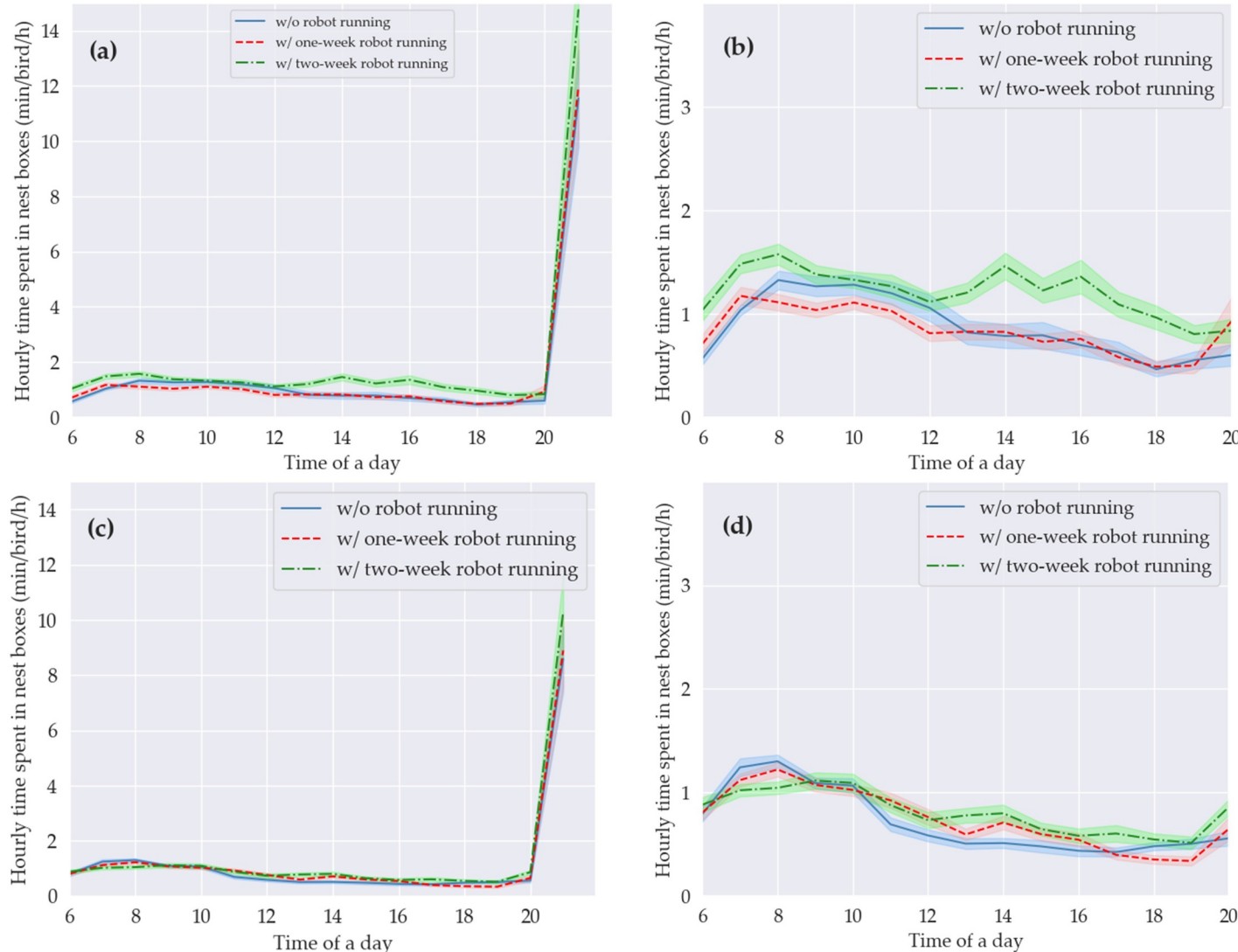

**Fig 6. Hourly time spent in nest boxes.** (a) From 6:00 to 21:00 in weeks 35–38; (b) from 6:00 to 20:00 in weeks 35–38; (c) from 6:00 to 21:00 in weeks 40–43; and (d) from 6:00 to 20:00 in weeks 40–43. Each value is the average of 4 values, and shading areas represent pooled standard errors of the means. 'w/' and 'w/o' represent with and without, respectively.

## Discussion

This research was aimed to investigate the effects of ground robot activity on hen floor egg reduction, production performance, stress response, bone quality, and nesting behaviors.

### Floor egg reduction

Overall floor egg rates in this study were significantly higher than typically reported rates (0.2–2% [4]), which could be attributed to differences in housing systems [6], genetics [26], and social interactions [27]. Birds used in this study were 34–43 weeks of age and likely to have already well-established laying patterns. Placing them into new pens around these periods could have disrupted laying patterns and caused high floor egg rates in this study [28]. Therefore, investigating robot impacts on birds having not begun laying may be better than those

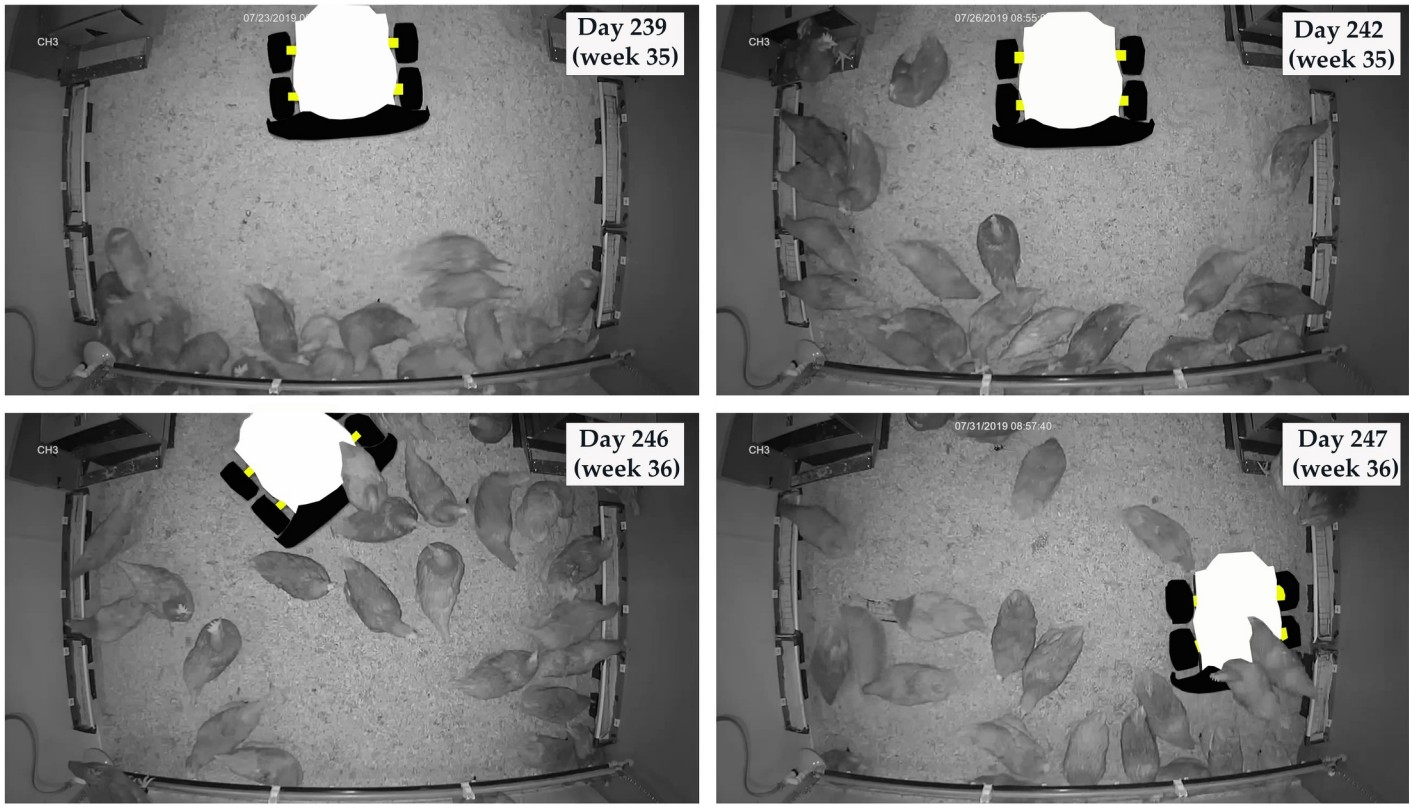

**Fig 7. Example pictures of hen-robot interactions for a pen with two-week robot running in phase 1 (weeks 35–38).**

with well-established patterns in commercial farms. Although high floor egg rates are not desired in commercial farms, they were favorable in this study for providing enough samples to measure the effects of robot operation.

Floor eggs were not significantly reduced overall, largely attributed to bird familiarization with the robot (Fig 7). Introducing other stimulations (e.g., varying robot speeds, sounds, vibration, and lights) may be considered for improving floor egg reduction via robot robots. Because of its relatively large size, the robot can only push birds in the open areas of the pen, but cannot reach birds hiding around corners, which can be secluded places for birds to lay eggs [29]. Smaller robots and reducing secluded areas are recommended for future applications. As a coincidence, the groups with robot treatments initially experienced higher floor egg rates than the control group, and negative hen laying behavior was potentially elicited by other floor-laying birds in the same pen [30]. Both of which may downgrade the effects of robot manipulations on reducing floor eggs, potentially resulting in insignificant floor egg reduction between the robot treatment and control groups.

Insufficient duration of robot manipulation may influence floor egg reduction. Weekly floor egg rates may be further decreased if the robot continued to run after the two-week operation period [11], according to the overall trend with the two-week robot running treatment. While such a data trend was detected during the experiments, the designated robot manipulation frequency could not be altered at an intermediate stage because of statistical results being required. A specific laying period (weeks 34–43) and small sample size (30) were tested because of limited time and resources. These factors may influence robot efficiency on floor egg reduction and should be considered for future robot testing in commercial farms.

Quantifying hen behavior adaptation to the robots was considered. Avoidance distance to the robot was used to describe hen- or broiler-robot interaction in open areas of commercial farms [11]. However, laying hens were placed in enclosed floor pens in this study and had restricted avoidance space even though they may have been fearful of the robot in the first several days, making the measure unsuitable for this study. Other behaviors, such as activity and restlessness, and fear responses may also be examined in the future for indicating bird adaptations to robots.

### Production performance, footpad health condition, and litter condition

Hen-day egg production was lower than the value provided in the Hy-Line Brown production manual [31], while feed intake and FCR were higher than levels indicated in the manual. Rearing environments (i.e., temperature, light intensity, light program, stocking density, and resource allocation) may not be the major reasons for the compromised production performance as they met the rearing requirements of the Hy-Line manual. Environmental variations during transportation could be the reason. Birds were transported at week 34 as recommended by farm managers, because younger birds may be more sensitive to environmental changes during transportation than older birds, and the complete social restructure and environment variations may cause physiological stress for young pullets and influence subsequent production performance [32, 33]. After arrival, birds gradually became acclimated to the new location and configuration of resources (e.g., feeder, drinker, and nest box), environmental conditions, and social hierarchies [34], and production performance was stabilized accordingly (Table 2). Fluctuations in production performance across bird ages may be attributed to human activities [35], such as weighing feed/chickens, collecting eggs, and adding feed. Overall, operating robots in hen houses may not introduce detrimental effects on hen production performance and could assist in farm management.

The robot treatments, however, significantly decreased egg mass. Ground robots may increase bird movement [10] and energy consumption and reduce nutrient deposition in eggs, accordingly resulting in lighter eggs. Similarly, Philippe et al. [36] also found lighter eggs produced by laying hens with higher activities in aviary systems than those in conventional cage and enriched colony systems. Interestingly, floor eggs were also lighter than nest eggs. Through live observation when animal caretakers fed birds, 1–2 subordinate birds in a pen could not use the nest boxes occupied by dominant birds [37]. They were also aggressively pecked by dominant birds once they appeared around feeders or other open areas. Therefore, the subordinate birds may have had insufficient feed, thus producing lighter eggs on the floor.

In this study, litter moisture contents were lower than previous studies (16.5% vs. 27.8–67.5% [15]), attributed to fresh litter provided for each flock and well-ventilated space. Lower litter moisture was not likely to induce footpad dermatitis [38], which was indicated by low footpad scores in this case.

### Stress response

The overall serum corticosterone concentrations were lower than those in a previous study (e.g., ~2000 pg/ml [39]). Concentration discrepancies may be attributed to various collection parts and time and extraction methods [39, 40]. Higher serum corticosterone concentrations in week 38 than week 34 may be attributed to housing and environment differences [41].

Previous studies have stated that ground robots did not introduce bird stress from behavior observation [11, 12], and the statement was confirmed from physiological responses of hens in this study. Again, birds may become acclimated to the presence of the robot gradually. As the ground robots may neither cause bird stress nor negatively influence production performance,

they could serve as acceptable precision animal management tools to assist producers and integrators in managing production systems. Besides floor egg reduction, other applications may include environmental monitoring, mortality removal, disinfection, feed distribution, bird migration, and litter scarification.

Measuring bird stress during or immediately following robot operations (Weeks 35–36) may help directly understand levels of bird stress caused by the robot. However, bird catching and handling during blood sampling have already caused bird stress, and blood sampling during or right after robot running (another potential stressor) may worsen the situation. The unexpected bird situation drove this project to select a safer time point for blood sampling two weeks after robot running. The insignificant corticosterone concentrations may indicate that birds may return to normal up to two weeks after robot running.

## Bone quality

Previous studies evaluated bone quality at the end of a production cycle (>72 weeks of bird age) [42, 43], therefore, there were few comparative results for bone quality of hens in weeks 34–43.

Although ground robots may encourage bird movement [10], they did not significantly improve hen bone quality (i.e., bone breaking force, fresh bone weight, dried bone weight, bone ash weight, and ash percentage). According to the manual of Hy-Line [31], Hy-Line Brown hens at week 30 are physically mature, in which cortical bone, medullary bone, digestive immune system, muscle, and reproductive tract are fully developed. In this case, bone quality of birds in weeks 34–43 was not likely to change significantly as long as they were provided with sufficient feed. The insignificant bone quality results may further provide scientific evidence that current settings (e.g., bird age and robot manipulation period) were not suitable for improving bird bone quality, and longer robot manipulation periods or using younger birds may be options.

## Coefficient of variations for stress response and bone quality

The CVs in this study reflect individual differences in physiological and skeletal responses. The robot did not induce significant differences in the CVs compared with other treatments. Again, birds may become accustomed to the robot, and, therefore, the robot may not be the major driver of individual variations. The overall CVs of bone quality were less than 30%, attributable to stabilized skeletal systems of birds after week 30 and similar as mentioned in Section 4.4. However, the CVs of stress response indicated by serum corticosterone concentrations ranged from 24.9% to 97.1% among treatment pens, which could be caused by social factors [44]. Per live observation, subordinate birds were subjected to restricted feed access by domain birds and were frequently pecked by dominant birds once presenting in open areas of a pen, leading to stress variations among birds and large CVs of serum corticosterone concentrations.

## Nesting behaviors

The daily time spent in nest boxes of this study was less than a previous study using the same RFID system (14.1 min vs. 63.7 min) [18], while both studies reported ~50% of daily time spent in nest boxes during the major laying period. Variation in housing systems (CF systems in the current study vs. enriched colony systems in the previous study) may be the major driver of time differences among studies. The birds in CF housing systems may spend time dust bathing, foraging, exploring, and ground pecking that cannot be performed in enriched cages [3],

thus reducing time spent in nest boxes. Changes in housing and environments for birds that had already been in their production cycle may also affect the nesting behaviors [28].

Although overall time spent in nest boxes was not significantly different, attributable to bird familiarization with the robot, the two-week robot running treatment indeed encouraged birds to use the nest boxes for a longer time numerically compared with other treatments in phase 1. One function of nest boxes is to provide the hens with egg laying space having little disturbance [40]. Therefore, birds may still prefer to stay in nest boxes to avoid robot disturbance despite habitation and familiarization with the robot.

The frequency distribution of number of birds simultaneously using a nest box was not significantly different among testing periods and robot treatments, which may not be the major factors influencing nesting space selection. Oliveira et al. [18] also reported that the cases of 0–5 birds simultaneously occupying the nest boxes took up the largest proportion out of all the cases. Perhaps, the hens may expect private and safe places to lay eggs [45]. As a result, even though the nest box is big enough to allow multiple hens to lay eggs together, the majority of the birds still chose to stay in the box singly.

Hens showed clear diurnal patterns of nesting behaviors either with or without robot running. The pattern from 6:00 to 13:00 coincided with the peak egg-laying periods [18]. However, the hens spent longer time numerically in nest boxes within the two hours before the lights were turned off. The difference was not from oviposition because the birds finished egg-laying during the morning. Mishra et al. [46] found no birds using nest boxes to lay eggs before the lights went off in an enriched colony system. Some birds may prefer a private and high place to rest and chose to rest in nest boxes overnight [47].

## Conclusions

Effects of ground robot operations on hen floor egg reduction, production performance, stress response, bone quality, and nesting behaviors were investigated. The results show that floor eggs were not effectively reduced during the whole experiment. The robot did not affect hen-day egg production, feed intake, feed conversion ratio, live body weight, serum corticosterone concentration, bone breaking force, ash percentage, and time spent in nest boxes. Additionally, birds with the two-week robot running treatment used nest boxes for longer time during the first two weeks of robot running than those with other treatments.

## Supporting information

**S1 Dataset.**
(DOCX)

## Acknowledgments

The authors wanted to appreciate the donation of chickens from our commercial partners and assistance from USDA technicians during experiment.

## Author Contributions

**Conceptualization:** Yang Zhao.

**Data curation:** Guoming Li, Xue Hui, Zach Porter, Sabin Poudel, Linan Jia, Bo Zhang.

**Formal analysis:** Guoming Li, Sabin Poudel, Linan Jia, Bo Zhang.

**Funding acquisition:** Wei Zhai, Joseph L. Purswell.

**Investigation:** Guoming Li, Zach Porter.

**Methodology:** Guoming Li, Yang Zhao, Wei Zhai, Joseph L. Purswell.

**Project administration:** Yang Zhao.

**Resources:** Yang Zhao.

**Software:** Guoming Li.

**Supervision:** Yang Zhao, Gary D. Chesser.

**Validation:** Guoming Li, Xue Hui.

**Visualization:** Guoming Li.

**Writing – original draft:** Guoming Li.

**Writing – review & editing:** Yang Zhao, Wei Zhai, Joseph L. Purswell, Gary D. Chesser.

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
