## [Decision Letter · Decision Letter 0]

21 Sep 2021

PONE-D-21-20853Effects of ground robots on hen floor egg reduction, production performance, stress response, bone quality, and behaviorPLOS ONE

Dear Dr. 

Thank you for submitting your manuscript to PLOS ONE. After careful consideration, we feel that it has merit but does not fully meet PLOS ONE’s publication criteria as it currently stands. Therefore, we invite you to submit a revised version of the manuscript that addresses the points raised during the review process.

Please submit your revised manuscript by Nov 05 2021 11:59PM. If you will need more time than this to complete your revisions, please reply to this message or contact the journal office at plosone@plos.org. Please include the following items when submitting your revised manuscript:A rebuttal letter that responds to each point raised by the academic editor and reviewer(s). You should upload this letter as a separate file labeled 'Response to Reviewers'.A marked-up copy of your manuscript that highlights changes made to the original version. You should upload this as a separate file labeled 'Revised Manuscript with Track Changes'.An unmarked version of your revised paper without tracked changes. You should upload this as a separate file labeled 'Manuscript'.

We look forward to receiving your revised manuscript.

Kind regards,

Balamuralikrishnan Balasubramanian

Academic Editor

PLOS ONE

Journal Requirements:

2. To comply with PLOS ONE submissions requirements, in your Methods section, please provide additional information on the animal research and ensure you have included details on (1) methods of sacrifice, (2) methods of anaesthesia and/or analgesia, and (3) efforts to alleviate suffering.

Y.Z.

N/A

Egg Industry Center

https://www.eggindustrycenter.org/

No

6. We note that Figures 1, 3 and 11 in your submission contain copyrighted images. All PLOS content is published under the Creative Commons Attribution License (CC BY 4.0), which means that the manuscript, images, and Supporting Information files will be freely available online, and any third party is permitted to access, download, copy, distribute, and use these materials in any way, even commercially, with proper attribution. For more information, see our copyright guidelines: http://journals.plos.org/plosone/s/licenses-and-copyright.

a. You may seek permission from the original copyright holder of Figures 1, 3 and 11 to publish the content specifically under the CC BY 4.0 license. 

Additional Editor Comments:

We have carefully evaluated your manuscript, We might, however, be able to accept it if you could respond adequately to the points that have been raised during the review process.

Reviewers' comments:

Reviewer's Responses to Questions

**Comments to the Author**

1. Is the manuscript technically sound, and do the data support the conclusions?

Reviewer #1: Partly

Reviewer #2: Yes

2. Has the statistical analysis been performed appropriately and rigorously? 

Reviewer #1: No

Reviewer #2: Yes

3. Have the authors made all data underlying the findings in their manuscript fully available?

Reviewer #1: No

Reviewer #2: Yes

4. Is the manuscript presented in an intelligible fashion and written in standard English?

Reviewer #1: Yes

Reviewer #2: Yes

5. Review Comments to the Author

Reviewer #1: This paper reports on the ability for a floor robot to prevent hens from floor laying with testing conducted in an experimental setting. This research is valuable to be able to determine if claims on commercially available products are indeed true and whether such devices do have positive impact on floor laying, without any accompanying negative impacts.

I do have some concerns about the low number of pen replicates (only 2 per treatment, which was increased by mixing birds part way through the trial and restricting nest box access). I also am concerned about the measures chosen for the hens and why they were selected. It seems like the study matches what may be assessed for broilers but laying hens are quite different and therefore what is indicative in broiler studies may be less informative for laying hen studies. For example, why was litter moisture content measured for a short-term trial, was the robot anticipated to affect litter moisture content? I can see it may have if the nest boxes were located in a tiered system and there was a lot more manure laid on the manure belt, but in the floor pen setting I’m not convinced of the link. Similarly, why only measure footpad dermatitis, why not other welfare measures in the WQ protocol. Particularly for such young hens where footpad dermatitis is not likely to be a major issue. And I’m not at all convinced of the need to measure bone quality in this study and I hope the birds were not euthanised just for this purpose, particularly when the authors interpret their lack of significant differences as expected given the age of the birds. Were all hens euthanised at the end of the trial rather than being rehomed and you took the opportunity to take samples, or did you euthanise hens to measure something that was predicted to show little impact anyway, that seems to be an ethical concern in the research. The hypothesis of increased activity I can see as being relevant for broilers, but not as relevant for hens in a floor setting where standing is likely to place just as much loading on the leg bones other activity. I cannot see that the robot activation for the short period of time would have substantial impact on activity and therefore bone strength, which is what your results show. In places the grammar and wording reads a bit awkward and I have flagged each sentence where I think the grammar needs to be checked through again. I have made other additional specific comments below.

Abstract

Line 28: This sentence was not clear to me until I read through the results that there was a reduction in all groups across time. I otherwise couldn’t understand why there was a decrease in floor laying in the control group. So maybe the sentence could start with ‘All floor eggs reduced across time with 18.9% and 34.0% reduction for the treatments…’

Line 29: here and elsewhere in the manuscript, ‘not obvious’ is a strange method of reporting. Do you mean ‘not significant’?

Line 43: ‘resources’ rather than ‘welfare enrichments’ would be more appropriate

Line 46: I prefer not to see ‘etc’ in scientific documents as it is not informative to the reader.

Line 55: what about nest box preference research such as Hunniford et al. 2018, AABS, 201 7-14.

Line 56: drawn attention in different fields.

Line 60: you state ‘poultry’ production, but if you mean just broilers then state this.

Line 61: what were the robots mentioned here meant to do?

Line 68: it seems redundant to mention that none of the aforementioned studies looked at floor eggs if they were all conducted on broiler growers who do not lay eggs. Could the sentence be reworded?

Lines 6973: I think these sentences could be reduced and made more succinct. You seem to double-state your aims. ‘Performance of floor eggs’ is awkward wording. Clarify specifically nesting behaviours were observed. Don’t use ‘etc’ in line 73, you either measured specific variables or you didn’t, ‘etc’ does not help the reader. The ordering of the sentences is not the clearest so please check through these few sentences.

Line 74: you state ‘primarily’ or were they only based on behavioural observations? And behavioural observations of what?

Line 77: I’m not convinced by the argument for bird activity, there are many papers on improved activity in loose housed systems versus caged systems. The reference included here is old and there have been so many other studies since, including review papers.

Line 80: Your reasoning for evaluating nesting behaviours is not quite clear from your statement. Obviously you want the birds to lay their eggs in the nest box and not on the floor, but what is the rationale for doing a detailed analyses of nesting behaviour versus just counting the eggs laid in the nest box?

Line 82: reduced floor eggs more.

Lines 82-85: The wording here is awkward and needs to be checked through again.

Line 90: what kind of housing system did the birds come from? 34 weeks seems quite old to obtain hens for this kind of study when they would already have established laying patterns, presumably they were already in peak lay when they were transported. Why did you not conduct the study with birds coming into lay and looking at the establishment of their nesting patterns from the beginning versus a disruption of nesting patterns?

Line 94: did you provide enough nest box space for 30 birds? What guidelines were followed for the choice of two nest boxes for 30 birds?

Line 100: What did you do with the camera footage. It is not clear here why you have set up cameras in the pens. Actually it only became somewhat clear when I read the discussion as I ‘think’ a lot of your reported anecdotal observations were based on camera footage? Unless they were live observations? In which case, what were the cameras set up for?

Line 115: how were the birds tagged?

Line 134: why did you measure litter moisture content? The rationale for this measurement is not clear to me. It was such a short trial with fresh litter placed at the beginning.

Line 137: why only footpad health when there are multiple measures in the WQ protocol. Did a single observer do the footpad scoring?

Lines 141-143: Why were only 6 birds selected? 12 birds/treatment. That is a very small number to detect treatment effects. Also, in what timeframe were the blood samples collected from each bird following bird pick-up? When in week 34 were the blood samples taken? Was this intended to be a baseline sample among treatment groups?

Lines 155 onwards: The rationale for taking the bones is not really clear to me. Were the birds killed specifically for this? The hypotheses for the bones are weak. The video cameras could have provided information about bird activity levels. Did you really predict significant differences from a few minutes of robot activation? I would think this measure would be more relevant to broilers who are otherwise very sedentary and go through rapid growth changes in short periods of time.

Line 155: at the end of the experiments.

Line 163-164: The wording is awkward in this sentence. ‘followed by Standard’

Line 168: what was the detection range for the RFID tags? Was this system validated to only record birds in the nest box rather than birds standing next to the nest box?

Line 169: how were the tags attached to the bird’s leg? You use the word ‘bonded’, were they glued on?

Line 178: In Equation 2

Line 182: why have the litter moisture and footpad quality measurements been included in this section on production performance?

Line 186: and the sum of both

Line 191: assessment protocol described earlier.

Line 197: you mention that the cort measurements were taken to reflect bird response to stress. What stressor? The week 34 measurement was presumably taken to measure response to the new housing?

Lines 205 and 206: just nesting behavior, not ‘nesting behavior responses’

Line 213: It would be helpful to remind the reader here of your n values. If your experimental unit was the pen, then how did you have a large enough sample size to conduct parametric statistics?

Line 218: did you control for multiple post-hoc comparisons?

Line 219: ‘were firstly added with a constant’, please check awkward grammar here.

Line 221: Please check the grammar here, awkward wording.

Line 222 onwards: What about the interaction effects? I cannot see where these were reported, were there no significant interactions?

Line 222: General result comment. I don’t see the value in repeating all results in tables and in figures. There are a very high number of tables/figures for the manuscript because many of the results are displayed twice.

Line 225: This is a very high rate of floor eggs, it may have been a better design to look at these robot impacts in birds beginning to lay rather than birds taken from a commercial facility after laying patterns were likely already established.

Line 228: without the robot running

Line 228: this is a strange result, clearly a lot of pen variation which is challenging when the pen replicates were so few. And if this is before the robot treatments actually started then this statement should be reworded to avoid confusion for the reader.

Line 231: reduced slowly or remained stable

Line 231: In the second phase of the experiments, the

Table 1 and elsewhere, it would be good to include test statistics rather than just the p-values.

Line 242: and shaded areas

Line 244: the wording here is awkward. What do you mean by ‘at least 2 pens have the robot treatment’. Do you mean ‘in which the robot treatment was implemented’? Applies to here and in other legends.

Line 253: parameters could change ? do you mean the parameters DID change?

Line 272: Data of egg mass are presented

Line 272: The eggs in weeks 40 to 43 were

Line 285: Data are from

Line 286: what does n = 24 refer to? It is not clear currently.

Line 304: in week 34 I presume this was taken as a baseline measurement so you would not predict the treatments to differ. I’m not quite sure why week 38 was selected for the second measure. By week 38 the birds had 2-3 weeks to adjust following the robots, if the robots had indeed caused stress. So what are you trying to answer by taking measurements at week 38? Are you assessing whether any effects on nest box use of the robot resulted in reduced cort measurements? If you wanted to look at shorter-term stress impacts of the robot then why were measurements not taken in the weeks the robots were running?

Line 308: I don’t think there is a need to state that blood samples were taken from the brachial vein here.

Line 350: and shaded area represent

Line 374: It would be good to start with a summary of the project and aims for reader clarity here

Line 376: strange wording: ‘higher than the regular ones’. Perhaps state ‘than typical reported rates’ instead.

Line 377: Don’t use ‘etc’ when you are listing explanations for differences, the reader does not know what ‘etc’ refers to so it is not informative.

Line 377: what about the fact that you took birds at 34 weeks of age when they likely already had well-established patterns of laying and you placed them into new pens. Wouldn’t that be a more likely explanation rather than standard explanations covering everything possible that can affect a bird.

Line 378: what do you mean by ‘not expected’. Do you mean ‘not desired’ instead? ‘Not expected’ does not flow with the rest of the sentence.

Line 382-383: Here and elsewhere you report on a lot of behavioural observations, but no other information in the manuscript on how these behavioural observations were conducted or where you are drawing this information from. You make a lot of interpretations based on what I presume are ‘anecdotal observations’, but yet they are quite informative to the reader and it may be good to include some more formal behavioural analysis in the results. The clear behavioural adaptation to the robot is valuable and more informative than measures such as litter moisture content, yet this was not studied in detail.

Line 386: I do not agree with the statement here and not sure what you mean by ‘regular enrichment’. I think the birds just adapted to the robot in the pen rather than it having some secondary effect on their fear levels which enabled them to adapt to the robot.

Line 387: Not sure the Bari et al. 2020 reference is appropriate to include here. I do not think it supports your point.

Line 391: These secluded places for birds to lay eggs is probably one of the primary issues in floor laying. So a robot that does not go into the secluded corners is going to be of far less value than the robots that do go into the secluded areas.

Line 392-393: Not sure I understand what this sentence means. ‘Insufficient frequency of robot…’

Line 399-400: What about a robot that ran when the birds are first starting to lay and getting used to the nest box to train them to not lay on the floor. What about a robot that vibrates when it senses a weight on top of it to try and prevent birds from jumping onto it?

Line 412: why was week 34 recommended? The birds are in peak production and have established patterns of laying. It is unclear why the birds were obtained at this age. Or did you purposely intend to test birds in peak production? In which case this needs to be made clearer in your aims because otherwise week 34 seems a strange age to obtain birds for a trial on egg laying.

Line 415: what about the complete social restructure and environment change? I would think a reduction in performance after that stressor would be expected, this seems more expected than unexpected.

Line 418: how do you know that the adaptation period alleviated physiological stress? How did you measure that?

Line 427: here is another case of the behavioural observations that are referred to as explanation for the results but there is no mention of how these observations were conducted. How did you observe subordinate and dominant birds? Was there not enough nest box space provided for 30 birds?

Line 430: how many subordinate birds did you have? Enough to change the egg mass results?

Line 431: when did the other studies assess their litter moisture content? Wasn’t their study across a flock cycle? Is it really comparable to this study?

Line 433: this is a broiler reference here. Do you have any laying hen references instead? Broiler behaviour/activity and welfare problems are different from those of laying hens. While both do get footpad dermatitis, the timelines are different and it would be good to see a laying hen reference included here instead for the litter contribution. Still not clear why only footpad dermatitis was included as a welfare measurement.

Line 447: what do you mean by ‘handling time differences’? And ‘individual variations’? Wasn’t it the same birds that you measured at 34 and 38 weeks?

Line 451: This is a broiler ref here, probably not applicable to a laying hen study with birds in floor pens. Even standing by laying hens can be greatly effective for increasing leg bone strength.

Line 455-456: With your concluding sentence here, why did you kill the birds for bone samples. I think the rationale is weak and ethically questionable.

Line 464: Again, you state ‘per observation’. What observations were conducted in this study? You are relying on them for a lot of interpretations yet they are not a main part of the aims and results.

Line 479: Was is it referred to as ‘the previous study’? Was it a comparative study that used the same RFID system?

Line 474: what about the change in housing for birds already in their production cycle? What about nest box space per bird? Did that differ between the two studies?

Line 477: for a longer time

Line 479: Not sure this reference is the best fit here. Is the conclusion then that the robot does have an impact on the birds’ behaviour?

Line 484: ‘did not like to share’. The wording is colloquial here. Please reword.

Line 489: ‘obviously’. Do you mean ‘significantly’?

Line 493: did the birds have perches in the commercial system they came from?

Line 550: check the ‘R’ which should be a registered trademark symbol

Reviewer #2: This in principle an interesting paper, as with the increasingly available technology for poultry farming, questions may arise to what extent they do have the intended effect and whether or not there are any side effects. Quite a number of measures have been included indicative of production and welfare. However, the number of replicates has been low and the setting was not comparable to commercial conditions. The latter received attention in the discussion, the first one not. Further, as a coincidence (I suppose), the groups with the robots started with more floor eggs than the control group, and although the proportional decrease is measured, it may have affected the results. Finally, it is not clear why the study has not been carried out from the begin of lay onwards, where floor eggs usually are a problem. So it also calls for further research. There are many tables and figures included, of which many provide similar information. The paper would benefit from reducing tables and figures, presenting the most important information, and removing the subheadings in the discussion section.

Some specific comments:

Line 52-53, restriction of litter access may have negative welfare consequences, please add

E.g., line 60, I think these references are not in the proper format

Line 75 as far as I understand, earlier studies were also based on broiler chickens. So therefore it might also be interesting to include laying hens, as has been done here

Line 82 should be ‘reduced floor eggs more’ I suppose

Line 83 I suppose nest box restriction was done to promote the occurrence of floor eggs, so that the effect could be measured.

It is nice to have many pictures, but one of the pen and one of the robot is sufficient

Resutls: please reduce number of figures and tables, especially where these present similar information. If there is no effect or difference, text might also be sufficient for some indicators, such as for foot pad health (all scores 0, no table needed). Eg. Mortality can be included in the text

Fig 11 remove from discussion section. If there is a figure it belongs to the results.

Line 394 and further, yes of course, resources can be limited but within the period of study, the robots could also have been present continuously. So why was this not done?

Line 404 and further can be shortened a lot not really relevant to the paper

Line 439 previous studies were on broilers, these on layers, I understood from your introductin

Line 448 and further, if bones are already mature, why then include this measure? Why not start at an earlier age with the trial?

6. PLOS authors have the option to publish the peer review history of their article (what does this mean?). If published, this will include your full peer review and any attached files.

Reviewer #1: No

Reviewer #2: No

---

## [Author Response · Author response to Decision Letter 0]

23 Nov 2021

Dear editor and reviewers,

We appreciated your insightful comments on improving the quality of our manuscript. We have tried our best to address the proposed questions, comments, and suggestions. The responses to specific comments are highlighted in blue font, and corresponding revisions can be found in the track-change manuscript. 

Editorial requirements:

Response 1: We have double checked the format requirements in the two attachments and reformatted previous inappropriate styles. Corresponding format changes can be found in the track-change manuscript.

2. To comply with PLOS ONE submissions requirements, in your Methods section, please provide additional information on the animal research and ensure you have included details on (1) methods of sacrifice, (2) methods of anaesthesia and/or analgesia, and (3) efforts to alleviate suffering.

Response 2: 1) There is no sacrifice involved in this study. 2) The anesthesia method was added, ‘At the end of each batch of experiment, birds were humanly euthanized according to the AVMA Guidelines [20].’ (L201-L202). 3) We only ran the robot in this study. Based on the data, although birds were unfamiliar with the robot at first but got used to it every soon. After that, we did not find severe suffering during the whole experiment and had no efforts to alleviate suffering.

3. Thank you for stating the following financial disclosure: ‘Y.Z., N/A, Egg Industry Center (https://www.eggindustrycenter.org/), No’. Please state what role the funders took in the study. If the funders had no role, please state: "The funders had no role in study design, data collection and analysis, decision to publish, or preparation of the manuscript." If this statement is not correct you must amend it as needed. Please include this amended Role of Funder statement in your cover letter; we will change the online submission form on your behalf.

Response 3: We added the statement ‘The funders (Egg Industry Center) had no role in study design, data collection and analysis, decision to publish, or preparation of the manuscript’ in the cover latter.

4. In your Data Availability statement, you have not specified where the minimal data set underlying the results described in your manuscript can be found. PLOS defines a study's minimal data set as the underlying data used to reach the conclusions drawn in the manuscript and any additional data required to replicate the reported study findings in their entirety. All PLOS journals require that the minimal data set be made fully available. For more information about our data policy, please see http://journals.plos.org/plosone/s/data-availability. Upon re-submitting your revised manuscript, please upload your study’s minimal underlying data set as either Supporting Information files or to a stable, public repository and include the relevant URLs, DOIs, or accession numbers within your revised cover letter. For a list of acceptable repositories, please see http://journals.plos.org/plosone/s/data-availability#loc-recommended-repositories. Any potentially identifying patient information must be fully anonymized. Important: If there are ethical or legal restrictions to sharing your data publicly, please explain these restrictions in detail. Please see our guidelines for more information on what we consider unacceptable restrictions to publicly sharing data: http://journals.plos.org/plosone/s/data-availability#loc-unacceptable-data-access-restrictions. Note that it is not acceptable for the authors to be the sole named individuals responsible for ensuring data access. We will update your Data Availability statement to reflect the information you provide in your cover letter.

Response 4: The minimal underlying data set was created for all tables and figures and uploaded as Supporting Information files in the submission system. The statement ‘The minimal underlining dataset is uploaded as Supporting Information files in the submission system.’ was added in the cover letter.

Response 5: We completed the animal use statement as follows. Hopefully, it is approved by the journal.

‘The animal use of this study was conducted according to the guidelines of the USDA-ARS Animal Care and Use Committee at the Mississippi State, Mississippi location (protocol 19-6 and date of approval 27 June 2019).’ (L127-L129)

6. We note that Figures 1, 3 and 11 in your submission contain copyrighted images. All PLOS content is published under the Creative Commons Attribution License (CC BY 4.0), which means that the manuscript, images, and Supporting Information files will be freely available online, and any third party is permitted to access, download, copy, distribute, and use these materials in any way, even commercially, with proper attribution. For more information, see our copyright guidelines: http://journals.plos.org/plosone/s/licenses-and-copyright. We require you to either (1) present written permission from the copyright holder to publish these figures specifically under the CC BY 4.0 license, or (2) remove the figures from your submission: a. You may seek permission from the original copyright holder of Figures 1, 3 and 11 to publish the content specifically under the CC BY 4.0 license. We recommend that you contact the original copyright holder with the Content Permission Form (http://journals.plos.org/plosone/s/file?id=7c09/content-permission-form.pdf) and the following text: “I request permission for the open-access journal PLOS ONE to publish XXX under the Creative Commons Attribution License (CCAL) CC BY 4.0 (http://creativecommons.org/licenses/by/4.0/). Please be aware that this license allows unrestricted use and distribution, even commercially, by third parties. Please reply and provide explicit written permission to publish XXX under a CC BY license and complete the attached form.” Please upload the completed Content Permission Form or other proof of granted permissions as an "Other" file with your submission. In the figure caption of the copyrighted figure, please include the following text: “Reprinted from [ref] under a CC BY license, with permission from [name of publisher], original copyright [original copyright year].” b. If you are unable to obtain permission from the original copyright holder to publish these figures under the CC BY 4.0 license or if the copyright holder’s requirements are incompatible with the CC BY 4.0 license, please either i) remove the figure or ii) supply a replacement figure that complies with the CC BY 4.0 license. Please check copyright information on all replacement figures and update the figure caption with source information. If applicable, please specify in the figure caption text when a figure is similar but not identical to the original image and is therefore for illustrative purposes only.

Response 6: We removed Fig. 3 previously, which contained duplicate and clear product labels. We also made the schematic drawing for the Figs. 1 and 7 (Fig. 11 previously), to avoid the copyright issue. We obtained the written permission for the modified Figs. 1 and 7, and the communication email is uploaded as well.

 

Comments to the Author

1. Is the manuscript technically sound, and do the data support the conclusions?

Reviewer #1: Partly

Reviewer #2: Yes

Response 7: We clarified the number (4) of replicate pens per treatment and some incorrect descriptions based on the reviewers’ comments. (L27 and L105)

2. Has the statistical analysis been performed appropriately and rigorously?

Reviewer #1: No

Reviewer #2: Yes

Response 8: We improved the statistical analysis and discussion based on the reviewers’ comments. Hopefully, the current manuscript can fit the reviewers’ requirement.

3. Have the authors made all data underlying the findings in their manuscript fully available?

Reviewer #1: No

Reviewer #2: Yes

Response 9: We supplemental the minimal underlying data set for all data tables and figures and uploaded as Supporting Information files in the submission system 

Reviewer #1

I do have some concerns about the low number of pen replicates (only 2 per treatment, which was increased by mixing birds part way through the trial and restricting nest box access). 

Response 10: We may not have expressed relevant contents appropriately that we had 4 pen replicates per treatment. ‘Two flocks of 180 Hy-Line Brown hens at week 34 were used, and birds in each flock were equally distributed into six CF pens, resulting in 4 pen replicates per treatment.’ (L26-L28) We believe the 4 pen replicates per treatment should be statistically sound.

I also am concerned about the measures chosen for the hens and why they were selected. It seems like the study matches what may be assessed for broilers but laying hens are quite different and therefore what is indicative in broiler studies may be less informative for laying hen studies. For example, why was litter moisture content measured for a short-term trial, was the robot anticipated to affect litter moisture content? I can see it may have if the nest boxes were located in a tiered system and there was a lot more manure laid on the manure belt, but in the floor pen setting I’m not convinced of the link. 

Response 11: Thanks for pointing these out. For the starter, we wanted to measure the robot effects on every possible aspect of laying hens in a preliminary study before starting large-scale experiments in commercial farms. Therefore, the floor pen was set for the preliminary examinations because of controlled environments, convenient management with limited labor, and so on. As the floor pens had many housing similarities to floor rearing systems of broilers, such as litter and drinking line, we borrowed some common indicators measuring broilers to test laying hens, hopefully providing some different ideas to hen industry. (L77-L84).

Indeed, we expected the robot could affect litter moisture content. The robot can go forward and backward and turn around within the pen, and the wheels may aerate litter via tumbling within these manipulations. Litter aeration may help to assist in drying the litter and reducing litter moisture content. The litter moisture content is not commonly measured in a short-term trial for laying hens, but previous studies (such as [20]) also measured significant differences of the parameter with various treatment for 34-week laying hens in an 8-week experiment. Therefore, we also expected some differences of litter moisture content in a 10-week experiment. (L166-L172)

Similarly, why only measure footpad dermatitis, why not other welfare measures in the WQ protocol. Particularly for such young hens where footpad dermatitis is not likely to be a major issue. 

Response 12: We had too many parameters measured at the end of the experiment, such as bone sampling, bird weighing, feed intake collection, litter sampling, egg counting and weighing, and footpad evaluation. We did want to measure more parameters but were short of hands at the end day. Therefore, we selected a representative parameter, footpad score, to reflect hen foot conditions, which may be potentially affected by ground robot running. It should be pointed out that we just wanted to understand the food conditions of hens in floor pens, which was also a primary concern for floor-rearing laying hens [21], and did not want to highlight welfare conditions of hens because poultry welfare is related to many aspects. Although young hens are not likely to have footpad dermatitis, Campbell et al. [22] reported a 0.3% footpad dermatitis rate for free-range hens at 20-36 weeks of age, which are similar to the bird ages in this study. While the housing systems were different among these two studies, ground robot running may potentially reduce bird contact with litter and result in better foot health. But such a hypothesis require to be measured in this study. (L173-L180)

And I’m not at all convinced of the need to measure bone quality in this study and I hope the birds were not euthanised just for this purpose, particularly when the authors interpret their lack of significant differences as expected given the age of the birds. Were all hens euthanised at the end of the trial rather than being rehomed and you took the opportunity to take samples, or did you euthanise hens to measure something that was predicted to show little impact anyway, that seems to be an ethical concern in the research. The hypothesis of increased activity I can see as being relevant for broilers, but not as relevant for hens in a floor setting where standing is likely to place just as much loading on the leg bones other activity. I cannot see that the robot activation for the short period of time would have substantial impact on activity and therefore bone strength, which is what your results show. 

Response 13: For the starter, the animal use of this study was conducted according to the guidelines of the USDA-ARS Animal Care and Use Committee at the Mississippi State, Mississippi location (protocol 19-6 and date of approval 27 June 2019). (L127-L129) The bone quality measurement and euthanasia methods were also included in the protocol and approved by the animal care and use committee, therefore, we do not have animal ethical problem in this study. All hens were euthanized at the end of this experiment, and tibia bone samples of the 10 tagged birds in each pen were scheduled to be collected at the same time. (L200-L201).

Secondly, from the two studies in our research team, we had observed that the robot can encourage movement and activity, such as walking and fleeing rather than just standing, for both broilers and hens. That is why we hypothesized that birds could perform those active behaviors in this study (L66-L70).

Finally, before this study, the robot effect on hens in cage-free systems, or specifically in floor pens, were a black box. We did not know how long or whether would the robot influence bird bone quality via encouraging bird movement. Now from this paper, we can deliver some information to practitioners that current settings (e.g., bird age, robot manipulation period, etc.) were not suitable for improving bird bone quality. Longer robot manipulation periods or using younger birds may be helpful for enhancing bird bone quality (L542-L547).

In places the grammar and wording reads a bit awkward and I have flagged each sentence where I think the grammar needs to be checked through again. I have made other additional specific comments below.

Response 14: We have corrected grammatical errors per the suggestions. The changes can be found in the track-change manuscript.

Abstract

Line 28: This sentence was not clear to me until I read through the results that there was a reduction in all groups across time. I otherwise couldn’t understand why there was a decrease in floor laying in the control group. So maybe the sentence could start with ‘All floor eggs reduced across time with 18.9% and 34.0% reduction for the treatments…’

Response 15: The sentence was changed to ‘…all floor eggs were reduced across time with 18.9% and 34.0% reduction for the treatments with robot operations…’ (L32-L33)

Line 29: here and elsewhere in the manuscript, ‘not obvious’ is a strange method of reporting. Do you mean ‘not significant’?

Response 16: all ‘obvious’ was changed to ‘significant’

Line 43: ‘resources’ rather than ‘welfare enrichments’ would be more appropriate

Response 17: ‘welfare enrichment’ was changed to ‘resources’. (L49).

Line 46: I prefer not to see ‘etc’ in scientific documents as it is not informative to the reader.

Response 18: all ‘etc’ was removed throughout the manuscript.

Line 55: what about nest box preference research such as Hunniford et al. 2018, AABS, 201 7-14.

Response 19: The reference ‘Hunniford, M. E., G. J. Mason, and T. M. Widowski. "Laying hens’ preferences for nest surface type are affected by enclosure." Applied Animal Behaviour Science 201 (2018): 7-14.’ was added as reference [8]. (L61)

Line 56: drawn attention in different fields.

Response 20: The sentence was changed to ‘…drawn attention in different fields…’ (L62)

Line 60: you state ‘poultry’ production, but if you mean just broilers then state this.

Response 21: These studies included both broiler and laying hens, for example, ‘Parajuli et al. [11] explored the broiler and hen responses…’ (L68)

Line 61: what were the robots mentioned here meant to do?

Response 22: The robot was running on litter floor to encourage bird usage of elevated platforms. (L67-L68)

Line 68: it seems redundant to mention that none of the aforementioned studies looked at floor eggs if they were all conducted on broiler growers who do not lay eggs. Could the sentence be reworded?

Response 23: The sentence was reworded to ‘These investigations have demonstrated critical values of robots to guide automation and precision management in poultry production, and robotics may have potential on reducing floor eggs of hens in CF housing systems but should be evaluated.’ (L72-L75)

Lines 69-73: I think these sentences could be reduced and made more succinct. You seem to double-state your aims. ‘Performance of floor eggs’ is awkward wording. Clarify specifically nesting behaviours were observed. Don’t use ‘etc’ in line 73, you either measured specific variables or you didn’t, ‘etc’ does not help the reader. The ordering of the sentences is not the clearest so please check through these few sentences.

Response 24: The relevant contents were removed.

 The influence of introducing robots on hen production performance (e.g., hen-day egg production, feed consumption, feed conversion ratio, etc.) was also examined. 

We removed all ‘etc’ from the manuscript and reworded relevant contents.

Line 74: you state ‘primarily’ or were they only based on behavioural observations? And behavioural observations of what?

Response 25: ‘primarily’ may be better, because in some studies with robots, production performance was also measured.

The sentence was changed to ‘they were primarily based on broilers and observation of behavior, such as bird avoidance distance to robot [12]’ (L86-L87)

Line 77: I’m not convinced by the argument for bird activity, there are many papers on improved activity in loose housed systems versus caged systems. The reference included here is old and there have been so many other studies since, including review papers.

Response 26: The old reference was removed, and two newer studies as follows were added. The relevant contents were revised to ‘…increasing activities in some loose housing system have been correlated to improving bird bone quality [3, 14]…’. (L89-L91)

Janczak, Andrew M., and Anja B. Riber. "Review of rearing-related factors affecting the welfare of laying hens." Poultry Science 94.7 (2015): 1454-1469.

Relić, Renata, et al. "Behavioral and health problems of poultry related to rearing systems." Ankara Üniversitesi Veteriner Fakültesi Dergisi 66.4 (2019): 423-428.

Line 80: Your reasoning for evaluating nesting behaviours is not quite clear from your statement. Obviously you want the birds to lay their eggs in the nest box and not on the floor, but what is the rationale for doing a detailed analyses of nesting behaviour versus just counting the eggs laid in the nest box?

Response 27: This is a very interesting question and discussed in the discussion section. “Counting the eggs laid in the nest box is helpful to understand general nest box utilization [45], but nest behavior evaluation with the RFID system in this study can provide detailed examination of bird using nest boxes interacted with robots. For example, whether birds used nest boxes more often during the period of robot running, how individual hens used nest boxes, and how many birds can use a nest box simultaneously. These measures are also critical for future bird flock management, nest box design, and robot application.” (L560-L564)

Line 82: reduced floor eggs more.

Response 28: The sentence was changed to ‘…longer robot running reduced floor eggs more…’ (L96).

Lines 82-85: The wording here is awkward and needs to be checked through again.

Response 29: The wordings were changed through the manuscript.

“Nest boxes were temporarily blocked during the experiment to simulate inadvertent events of nest box restriction in commercial farms, and the nest box restriction may also promote the occurrence of floor eggs, so that the effect could be measured.” (L96-L99)

“Phase 2 (40-43 weeks of bird age) was designed to investigate the floor egg reduction via robot after nest box restriction” (L144-L145).

Line 90: what kind of housing system did the birds come from? 34 weeks seems quite old to obtain hens for this kind of study when they would already have established laying patterns, presumably they were already in peak lay when they were transported. Why did you not conduct the study with birds coming into lay and looking at the establishment of their nesting patterns from the beginning versus a disruption of nesting patterns?

Response 30: The birds came from a commercial farm with aviary housing systems (L106). 

We did want to test younger birds at first. However, farm managers recommended us to transport older birds since they get used to new environment faster and maintain better production performance. Younger birds could be sensitive to environment changes during transportation, and inconsistent environments among the original farm, transportation truck, and experimental house may cause physiological stress for birds and influence subsequent production performance. (L483-L487) We wanted to balance various aspects in this comprehensive project, and younger birds can be further researched in the future. 

We should point out that conducting such a comprehensive project is not easy, and we need to balance many aspects. We did want to measure young cage-free birds at first. However, Mississippi State does not have cage-free housing farms that can meet our requirement, and we need to seek the collaboration from the nearest state, Kentucky State. We need to drive around 6 hours to pick up birds, and based on the biosecurity protocol in the farm, we also need to leave the farm by 4:00 am, at which the weather was cold during winter. Considering cold stress, transportation stress, and adaptation to new housing, the experienced farm managers suggested us to use older birds that may more likely to overcome these challenges and recover faster after transportation than younger birds. 

Line 94: did you provide enough nest box space for 30 birds? What guidelines were followed for the choice of two nest boxes for 30 birds?

Response 31: The 30 birds were provided with two nest boxes (60 cm long, 53 cm wide, and 53 cm high for each). The nest box space allowance was 212 cm2/hen that was larger than the 86 cm2/hen nest space in commercial aviary systems and 62 cm2/hen in enrich colony systems [17], and thus may be sufficient for hens in this study (L111-L113).

Line 100: What did you do with the camera footage. It is not clear here why you have set up cameras in the pens. Actually it only became somewhat clear when I read the discussion as I ‘think’ a lot of your reported anecdotal observations were based on camera footage? Unless they were live observations? In which case, what were the cameras set up for?

Response 32: The cameras were set up for creating a relatively enclosed and safe environment. The six pens were in two enclosed rooms (three for each room), and there was a passageway in the middle of the two rooms in the same house. (L107-L109) The video recorder and monitors showing what the cameras captured were placed in the passageway. Human stayed in the passageway when the robot was roaming in a pen to avoid human interference and watched the situations within the pen through cameras, so that any inappropriate events (e.g., severe bird stress caused by the robot) can be stopped timely. (L118-L121)

Providing evidences of bird adaption to robot is another purpose of the recorded images, but is not the initial purpose that the camera were set up for. 

Line 115: how were the birds tagged?

Response 33: Ten birds in each pen with two radio frequency identification (RFID) tags on one leg were used for blood sampling and bone sampling. (L142-L144)

Line 134: why did you measure litter moisture content? The rationale for this measurement is not clear to me. It was such a short trial with fresh litter placed at the beginning.

Response 34: For the starter, we wanted to measure the robot effects on every possible aspect of laying hens in a preliminary study before starting large-scale experiments in commercial farms. Therefore, the floor pen was set for the preliminary examinations because of controlled environments, convenient management with limited labor, and so on. As the floor pens had many housing similarities to floor rearing systems of broilers, such as litter and drinking line, we borrowed some common indicators measuring broilers to test laying hens, hopefully providing some different ideas to hen industry. (L77-L84).

Indeed, we expected the robot could affect litter moisture content. The robot can go forward and backward and turn around within the pen, and the wheels may aerate litter via tumbling within these manipulations. Litter aeration may help to assist in drying the litter and reducing litter moisture content. The litter moisture content is not commonly measured in a short-term trial for laying hens, but previous studies (such as [20]) also measured significant differences of the parameter with various treatment for 34-week laying hens in an 8-week experiment. Therefore, we also expected some differences of litter moisture content in a 10-week experiment. (L167-L173)

Line 137: why only footpad health when there are multiple measures in the WQ protocol. Did a single observer do the footpad scoring?

Response 35: We had too many parameters measured at the end of the experiment, such as bone sampling, bird weighing, feed intake collection, litter sampling, egg counting and weighing, and footpad evaluation. We did want to measure more parameters but were short of hands at the end day. Therefore, we selected a representative parameter, footpad score, to reflect hen foot conditions, which may be potentially affected by ground robot running. It should be pointed out that we just wanted to understand the food conditions of hens in floor pens, which was also a primary concern for floor-rearing laying hens [21], and did not want to highlight welfare conditions of hens because poultry welfare is related to many aspects. (L174-L179)

The footpad scoring was done by two trained observers, and evaluation results were mutually verified. (L163-L165)

Lines 141-143: Why were only 6 birds selected? 12 birds/treatment. That is a very small number to detect treatment effects. Also, in what timeframe were the blood samples collected from each bird following bird pick-up? When in week 34E were the blood samples taken? Was this intended to be a baseline sample among treatment groups?

Response 36: The number of birds for blood sampling = 6 birds/pen/flock × 2 pen replicates per treatment in a flock × 2 flocks = 24 birds/treatment (L187). 6 birds out of 30 birds per pen (20%) were selected for blood sampling, and this number was recommended by the poultry science experts from the poultry science department of Mississippi State University. Too many sampled birds can introduce too much human interference and may downgrade overall production performance, while too few sampled birds can result in insufficient data for covering the whole pen situation. 

We transported chickens at the start of week 34, and then after several days of acclimation (usually at the end of week 34), poultry science technicians observed bird health status to determine the feasibility of blood sampling. The blood samples during this period was used as baseline samples among treatment groups (L183-L185). 

Lines 155 onwards: The rationale for taking the bones is not really clear to me. Were the birds killed specifically for this? The hypotheses for the bones are weak. The video cameras could have provided information about bird activity levels. Did you really predict significant differences from a few minutes of robot activation? I would think this measure would be more relevant to broilers who are otherwise very sedentary and go through rapid growth changes in short periods of time.

Response 37: 

1. The birds were scheduled to be euthanized for the bone testing, which was approved by the animal use committee at Mississippi State. (L127-L129)

2. Although video cameras can provide information about bird activity levels, but bird bone quality with the activity remains unclear if we only examined the video data. 

3. We had expectation on the significant difference of bone quality affected by robot running. The total robot running time per week = 5 min per half hour × 2 halves per hour × 7 hours per day (from 7:00 to 13:00) × 5 days per week (from Monday to Friday) = 350 min per week. (L138-L140) This number was not small, and the time of robot running for influence bird bone quality remains to be explored. 

4. The bone quality was indeed typically measured for broilers that grow fast in short periods of time. However, in this study, laying hens were placed in floor pens, and they have more chances to run and walk. Meanwhile, their activity may also be encouraged by robot running. Therefore, we still expected some difference on bird bone quality when we started to make the experimental plan.

Line 155: at the end of the experiments.

Response 38: The sentence was changed to “At the end of each flock of experiments…” (L201).

Line 163-164: The wording is awkward in this sentence. ‘followed by Standard’

Response 39: The sentence was changed to “All procedures were followed as described…” (L210-L211).

Line 168: what was the detection range for the RFID tags? Was this system validated to only record birds in the nest box rather than birds standing next to the nest box?

Response 40: 

1. The RFID tag is a passive sensor, which receives signals rather than emitting signals. It does not have detection range. (L216)

2. Yes. Before birds entered the pens, we adjusted the power of the RFID antennas, so that the system only record the birds in the nest box and excluded the birds out of the nest box. (L218-L219)

Line 169: how were the tags attached to the bird’s leg? You use the word ‘bonded’, were they glued on?

Response 41: The RFID tags were bonded onto legs of birds using zip ties (L217-L218).

Line 178: In Equation 2

Response 42: It was changed to ‘In Equation 2’ (L228).

Line 182: why have the litter moisture and footpad quality measurements been included in this section on production performance?

Response 43: The section title was changed to “Production performance, foot health condition, litter condition” (L155).

Line 186: and the sum of both

Response 44: It was changed to “and the sum of both” (L236).

Line 191: assessment protocol described earlier.

Response 45: It was changed to “the assessment protocol described earlier” (L241)

Line 197: you mention that the cort measurements were taken to reflect bird response to stress. What stressor? The week 34 measurement was presumably taken to measure response to the new housing?

Response 46: The stressor was presumably the robot. (L248-L249). The week 34 measurement was taken to build the baseline serum corticosterone concentration when birds adapted to the new housing. (L250-L251)

Lines 205 and 206: just nesting behavior, not ‘nesting behavior responses’

Response 47: “nesting behavior responses” were changed to “nesting behaviors” (L258-L263)

Line 213: It would be helpful to remind the reader here of your n values. If your experimental unit was the pen, then how did you have a large enough sample size to conduct parametric statistics?

Response 48: 

1. n values were supplemented. “The sample size (n) with robot treatments was 40 for weekly floor egg rate, hen-day egg production, feed intake, FCR, and time spent in nest boxes, 32 for relative floor egg reduction, 24 for nest egg mass and floor egg mass, 4 for live body weight, foot pad score, and mortality, 6 for litter moisture content, 8 for serum corticosterone concentrations, bone breaking force, fresh bone weight, dried bone weight, bone ash weight, and ash percentage.” (L277-L281)

2. The experimental unit was the pen, and we had 4 pen replicates per treatments. Some measures included those in different weeks to increase enough sample sizes of statistical analysis. 

Line 218: did you control for multiple post-hoc comparisons?

Response 49: Yes, we used Fisher’s least significant difference method for multiple post-hoc comparisons. (L272)

Line 219: ‘were firstly added with a constant’, please check awkward grammar here.

Response 50: The sentence was changed to “A constant of 1 was added to all percentage data in decimal form to eliminate negatives” (L273).

Line 221: Please check the grammar here, awkward wording.

Response 51: The sentence was changed to “The logarithm transform prior to statistical analysis made the percentage data fit to normal distribution as much as possible, so that the statistical analysis from the data became more valid.” (L275-L277)

Line 222 onwards: What about the interaction effects? I cannot see where these were reported, were there no significant interactions?

Response 52: Yes. There was no interaction effects for all measures. “Interaction effects of robot treatment and bird age were not observed for all measures (P≥0.05). Therefore, detailed statistical results about interaction effects were not reported in the following tables and figures.” (L283-L286)

Line 222: General result comment. I don’t see the value in repeating all results in tables and in figures. There are a very high number of tables/figures for the manuscript because many of the results are displayed twice.

Response 53: We removed previous figs 4, 5, and 8, which had some duplicate contents compared to corresponding tables. Changes can be found in the track-change manuscript. 

Line 225: This is a very high rate of floor eggs, it may have been a better design to look at these robot impacts in birds beginning to lay rather than birds taken from a commercial facility after laying patterns were likely already established.

Response 54: Thank you for point this out. We placed these contents in the discussion section. “The high floor eggs also indicate that it may have been a better design to look at these robot impacts in birds beginning to lay rather than birds taken from a commercial facility after laying patterns were likely already established.” (L431-L433)

Line 228: without the robot running

Response 55: Relevant contents were deleted.

The initial weekly floor egg rates in the pens without the robot running were nearly 30% lower than those with robot treatments.

Line 228: this is a strange result, clearly a lot of pen variation which is challenging when the pen replicates were so few. And if this is before the robot treatments actually started then this statement should be reworded to avoid confusion for the reader.

Response 56: Relevant contents were deleted to avoid confusion. 

The initial weekly floor egg rates in the pens without the robot running were nearly 30% lower than those with robot treatments.

Line 231: reduced slowly or remained stable

Response 57: The sentence was changed to “the floor eggs were reduced slowly among pens” (L293)

Line 231: In the second phase of the experiments, the Table 1 and elsewhere, it would be good to include test statistics rather than just the p-values.

Response 58: We did not separate the analysis into two phases. The data with robot treatments and at different ages was analyzed with two-way ANOVA. Considering most of the parameters were not significantly different among robot treatments, the p-values may be enough and succinct to describe the story. We also supplemented test statistic information in the Minimal Data Set.

Line 242: and shaded areas

Response 59: The figure was deleted, and there is no “and shaded areas” now.

Line 244: the wording here is awkward. What do you mean by ‘at least 2 pens have the robot treatment’. Do you mean ‘in which the robot treatment was implemented’? Applies to here and in other legends.

Response 60: The figure was removed, and there is no “at least 2 pens have the robot treatment” now.

Line 253: parameters could change ? do you mean the parameters DID change?

Response 61: The sentence was changed to “The three parameters changed with bird ages (P<0.01).” (L308-L309)

Line 272: Data of egg mass are presented

Response 62: The sentence was changed to “Data of egg mass are presented” (L327)

Line 272: The eggs in weeks 40 to 43 were

Response 63: The sentence was changed to “The eggs in weeks 40 to 43 were” (L327)

Line 285: Data are from

Response 64: It was changed to “Data are from” (L336)

Line 286: what does n = 24 refer to? It is not clear currently.

Response 65: Egg mass was measured from weeks 40 to 43 in flock 2. (L268-L269). Therefore, n= 1 measure per week per pen × 4 week × 6 pens = 24

Line 304: in week 34 I presume this was taken as a baseline measurement so you would not predict the treatments to differ. I’m not quite sure why week 38 was selected for the second measure. By week 38 the birds had 2-3 weeks to adjust following the robots, if the robots had indeed caused stress. So what are you trying to answer by taking measurements at week 38? Are you assessing whether any effects on nest box use of the robot resulted in reduced cort measurements? If you wanted to look at shorter-term stress impacts of the robot then why were measurements not taken in the weeks the robots were running?

Response 66: This was a very arguable point when we started to make the experimental plan. From one aspect, we wanted to measured bird stress levels during robot running or right after robot running, which can examine whether robot running in floor pens would cause bird stress. However, blood sampling for measuring serum corticosterone concentrations requires bird catching and handling, which could also cause bird stress. Additionally, we did not have too much experience on robot running for laying hens in enclosed floor pens, and we worried about the blood sampling during robot running or right after robot running may cause unexpectedly severe bird stress. Therefore, we decided to leave two weeks after robot running for birds to calm down. Our results also demonstrated that birds returned to be normal for maximum two weeks after robot running based on the insignificant results among the treatments. We added these contents in the discussion section. (L524-L533)

Line 308: I don’t think there is a need to state that blood samples were taken from the brachial vein here.

Response 67: The content was removed.

The blood sampling parts were brachial veins.

Line 350: and shaded area represent

Response 68: The figure was removed, and there is no “and shaded area represent” now.

Line 374: It would be good to start with a summary of the project and aims for reader clarity here

Response 69: The summary was added. “This research was mainly aimed to investigate ground robot effects on hen floor egg reduction. Production performance, stress response, bone quality, and nesting behaviors were also measured to test whether the ground robot running would cause side effects on these.” (L421-L423)

Line 376: strange wording: ‘higher than the regular ones’. Perhaps state ‘than typical reported rates’ instead.

Response 70: It was changed to “than typical reported rates” (L425)

Line 377: Don’t use ‘etc’ when you are listing explanations for differences, the reader does not know what ‘etc’ refers to so it is not informative.

Response 71: All “etc” were removed from the manuscript.

Line 377: what about the fact that you took birds at 34 weeks of age when they likely already had well-established patterns of laying and you placed them into new pens. Wouldn’t that be a more likely explanation rather than standard explanations covering everything possible that can affect a bird.

Response 72: Thanks for the valuable suggestion. We added this into the discussion. “Birds used in this study were 34-43 weeks of age when they already had well-established patterns of laying. And placing them into new pens around these periods may disrupt the established laying pattern could also be the reason of high floor egg rates in this study [29].” (L427-L429)

Line 378: what do you mean by ‘not expected’. Do you mean ‘not desired’ instead? ‘Not expected’ does not flow with the rest of the sentence.

Response 73: It was changed to “not desired” (L430)

Line 382-383: Here and elsewhere you report on a lot of behavioural observations, but no other information in the manuscript on how these behavioural observations were conducted or where you are drawing this information from. You make a lot of interpretations based on what I presume are ‘anecdotal observations’, but yet they are quite informative to the reader and it may be good to include some more formal behavioural analysis in the results. The clear behavioural adaptation to the robot is valuable and more informative than measures such as litter moisture content, yet this was not studied in detail.

Response 74: This is a very tough question. We also thought about quantifying hen behavior adaptation to the robots. However, we cannot find appropriate behavior measures to describe the adaptation. Avoidance distance to the robot was the one our team used to describe hen- or broiler-robot interaction in open areas of commercial farms [11]. However, laying hens were placed enclosed floor pens and had limited space to run away even though they may have been scared by the robot in the first several days. Therefore, the avoidance distance measure was not suitable in this study. Meanwhile, because of unexpected events, we lost the video day in the first flock, resulting in insufficient data for the statistical analysis. As a result, we only can present the anecdotal observations in the discussion section rather than the quantitative observations in the result section. This can be a future work of robot application in egg industry. (L462-L470)

Line 386: I do not agree with the statement here and not sure what you mean by ‘regular enrichment’. I think the birds just adapted to the robot in the pen rather than it having some secondary effect on their fear levels which enabled them to adapt to the robot.

Response 75: The statement was removed.

Once birds became accustomed to the robot, it may be treated as a regular enrichment and thus reduced fear levels and subsequent avoidance behaviors [13, 31].

Line 387: Not sure the Bari et al. 2020 reference is appropriate to include here. I do not think it supports your point.

Response 76: The relevant content was removed.

Once birds became accustomed to the robot, it may be treated as a regular enrichment and thus reduced fear levels and subsequent avoidance behaviors [13, 31].

Line 391: These secluded places for birds to lay eggs is probably one of the primary issues in floor laying. So a robot that does not go into the secluded corners is going to be of far less value than the robots that do go into the secluded areas.

Response 77: We reworded the sentence “Smaller robots that can reach secluded areas (for birds) and reducing secluded areas are recommended for future applications”. (L444-L445)

Line 392-393: Not sure I understand what this sentence means. ‘Insufficient frequency of robot…’

Response 78: It was changed to “Insufficient duration of robot manipulation”. (L445)

Line 399-400: What about a robot that ran when the birds are first starting to lay and getting used to the nest box to train them to not lay on the floor. What about a robot that vibrates when it senses a weight on top of it to try and prevent birds from jumping onto it?

Response 79: Thanks for providing such insightful points. We added these in the discussion section. “Other possibilities can also be investigated in the future to reduce floor eggs, for example, a robot starts to run when the birds are first starting to lay and getting used to the nest box to train them not to lay on the floor, and a robot vibrates when it senses a weight on top of it to try and prevent birds from jumping onto it.” (L456-L459)

Line 412: why was week 34 recommended? The birds are in peak production and have established patterns of laying. It is unclear why the birds were obtained at this age. Or did you purposely intend to test birds in peak production? In which case this needs to be made clearer in your aims because otherwise week 34 seems a strange age to obtain birds for a trial on egg laying.

Response 80: Birds were transported at week 34 as recommended by farm managers because younger birds may not be able to survive and maintain good production performance after transportation stress. (L484-L485).

We should point out that conducting such a comprehensive project is not easy, and we need to balance many aspects. We did want to measure young cage-free birds at first. However, Mississippi State does not have cage-free housing farms that can meet our requirement, and we need to seek the collaboration from the nearest state, Kentucky State. We need to drive around 6 hours to pick up birds, and based on the biosecurity protocol in the farm, we also need to leave the farm by 4:00 am, at which the weather was cold during winter. Considering cold stress, transportation stress, and adaptation to new housing, the experienced farm managers suggested us to use older birds that may more likely to overcome these challenges and recover faster after transportation than younger birds. 

Line 415: what about the complete social restructure and environment change? I would think a reduction in performance after that stressor would be expected, this seems more expected than unexpected.

Response 81: The sentence was changed to “and the complete social restructure and environment changes may cause physiological stress for birds and influence subsequent production performance” (L486-L487)

Line 418: how do you know that the adaptation period alleviated physiological stress? How did you measure that?

Response 82: We cannot measure that. Relevant contents were removed to avoid confusion.

The acclimation period alleviated physiological stress.

Line 427: here is another case of the behavioural observations that are referred to as explanation for the results but there is no mention of how these observations were conducted. How did you observe subordinate and dominant birds? Was there not enough nest box space provided for 30 birds?

Response 83: When the animal caretakers did the daily chores (e.g., feeding birds), they observed bird situations in a pen and recorded unusual events (e.g., bird pecking) in the corresponding spreadsheet. (L123-L125). That is where the behavioral observations came from. The subordinate and dominant birds can be judged through observing pecking behavior in the pens. 

The nest box space provided for 30 birds was 212 cm2/hen that was larger than the 86 cm2/hen in commercial aviary systems and 62 cm2/hen in enrich colony systems [18], and thus may be sufficient for hens in this study. (L111-L113)

Line 430: how many subordinate birds did you have? Enough to change the egg mass results?

Response 84: There were 1-2 subordinate birds in a pen. (L501) The floor egg rate was 31.6%-59.6% in this case and hen-day egg production was 77.3-90.9%, converting to 7-16 floor eggs for 30 birds daily. And 1-2 floor eggs produced by the subordinate birds occupied 6.3%-28.5% of the total floor eggs. We thought such a ratio may influence the overall floor egg mass results. In addition, we just wanted to discuss one possibility to cause the lighter weight of floor eggs compared to nest eggs. That is why we used uncertainty in this sentence “subordinate birds may not have sufficient feed and produce lighter eggs on floor” (L502-L503).

Line 431: when did the other studies assess their litter moisture content? Wasn’t their study across a flock cycle? Is it really comparable to this study?

Response 85: The experiment was conducted from 17 to 76 weeks of bird age. The reviewer is right that the following reference is not comparable to this study and removed.

Oliveira JL, Xin H, Chai L, Millman ST. Effects of litter floor access and inclusion of experienced hens in aviary housing on floor eggs, litter condition, air quality, and hen welfare. Poult Sci. 2019;98(4):1664-77. doi: https://doi.org/10.3382/ps/pey525.

The following reference was added. It is for 34- to 42-week-old laying hens in floor pens.

Kang, H. K., et al. "Effects of stock density on the laying performance, blood parameter, corticosterone, litter quality, gas emission and bone mineral density of laying hens in floor pens." Poultry Science 95.12 (2016): 2764-2770.

Line 433: this is a broiler reference here. Do you have any laying hen references instead? Broiler behaviour/activity and welfare problems are different from those of laying hens. While both do get footpad dermatitis, the timelines are different and it would be good to see a laying hen reference included here instead for the litter contribution. Still not clear why only footpad dermatitis was included as a welfare measurement.

Response 86: The broiler reference was removed.

De Jong IC, Gunnink H, Van Harn J. Wet litter not only induces footpad dermatitis but also reduces overall welfare, technical performance, and carcass yield in broiler chickens. J Appl Poult Res. 2014;23(1):51-8. doi: https://doi.org/10.3382/japr.2013-00803.

And the laying hen study was added.

Wang, G., C. Ekstrand, and J. Svedberg. "Wet litter and perches as risk factors for the development of foot pad dermatitis in floor-housed hens." British Poultry Science 39.2 (1998): 191-197.

We measured footpad dermatitis for evaluating hen foot heath rather than assessing animal welfare. The rationale of this measure can also be found in the above responses. 

Line 447: what do you mean by ‘handling time differences’? And ‘individual variations’? Wasn’t it the same birds that you measured at 34 and 38 weeks?

Response 87: The “handling time differences” and “individual variations” were not appropriate to describe the data trend. The content was changed to “Higher serum corticosterone concentrations in week 38 than that in week 34 may be attributed to housing and environment differences [41]” (L522-L523)

Line 451: This is a broiler ref here, probably not applicable to a laying hen study with birds in floor pens. Even standing by laying hens can be greatly effective for increasing leg bone strength.

Response 88: This is the only reference that we can find to support the statement “ground robots may encourage more bird movement”, which we did observe in the first several days of robot running through recorded videos. We would kindly ask for maintaining this reference in the discussion, otherwise, we may discuss that statement with the “anecdotal observations”. 

Line 455-456: With your concluding sentence here, why did you kill the birds for bone samples. I think the rationale is weak and ethically questionable.

Response 89: We should point out that these contents are the interpretation of the results rather than the prior knowledge that we had already known the robot effects on hen bone quality before the experiments and results. Kindly, the reviewers are recommended to find detailed rationale explanation on bone quality measure in the above responses. 

Line 464: Again, you state ‘per observation’. What observations were conducted in this study? You are relying on them for a lot of interpretations yet they are not a main part of the aims and results.

Response 90: When the animal caretakers did the daily chores (e.g., feeding birds), they observed bird situations in a pen and recorded unusual events (e.g., bird pecking) in the corresponding spreadsheet. (L124-L125). That is where the behavioral observations came from. We would think that these recorded anecdotal events are quite valuable and supportive on understanding the results, and we may not have to set up another main part of the aims and results for understanding current results.

Line 479: Was is it referred to as ‘the previous study’? Was it a comparative study that used the same RFID system?

Response 91: We are confused by this question. In the previous Line 479, there was only “boxes for longer time to avoid it [35].”, and we cannot find “the previous study” and “RFID” system here. If the reviewers can further clarify the question, we would love to address the question as well.

35. Cronin GM, Hemsworth PH. The importance of pre-laying behaviour and nest boxes for laying hen welfare: a review. Animal Production Science. 2012;52(7):398-405. doi: https://doi.org/10.1071/AN11258.

Line 474: what about the change in housing for birds already in their production cycle? What about nest box space per bird? Did that differ between the two studies?

Response 92: We added the discussion “Changes in housing and environments for birds that had been already in their production cycle may also affect the nesting behaviors [29].” (L571-L573). The nest box space allowance was 212 cm2/hen that was larger than the 86 cm2/hen nest space in commercial aviary systems and 62 cm2/hen in enrich colony systems [18]. (L111-L113). The reference [3] is a review paper rather than a research paper. 

Line 477: for a longer time

Response 93: It was changed to “for a longer time” (L576).

Line 479: Not sure this reference is the best fit here. Is the conclusion then that the robot does have an impact on the birds’ behaviour?

Response 94: The reference is not related to robot. We just wanted to support the statement with the reference that the nest box may provide birds with space without disturbance. The relevant content was changed to “One function of nest boxes might be to provide the hens that choose to lay there with a location where they are less disturbed before egg laying [40]. Therefore, despite habitation to the robot, birds may still be disturbed by the robot and stay in nest boxes for a longer time to avoid it.” (L576-L578)

Line 484: ‘did not like to share’. The wording is colloquial here. Please reword.

Response 95: The relevant content was removed.

and did not like to share nesting space with others

Line 489: ‘obviously’. Do you mean ‘significantly’?

Response 96: It was changed to “numerically”. “the hens spent numerically longer time in nest boxes” (L588-L589)

Line 493: did the birds have perches in the commercial system they came from?

Response 97: The nest box has a perch in front of the entry (L113-L114), and this perch is just for small floor pen system rather than commercial farm systems.

Line 550: check the ‘R’ which should be a registered trademark symbol

Response 98: It was changed to “®”

Reviewer #2

However, the number of replicates has been low and the setting was not comparable to commercial conditions. The latter received attention in the discussion, the first one not. 

Response 99: We had two flocks of birds, each having 180 birds distributed into 6 pens. Therefore, we had 4 pen replicates per treatment, which is typical in poultry experiments (L25-L28)

Further, as a coincidence (I suppose), the groups with the robots started with more floor eggs than the control group, and although the proportional decrease is measured, it may have affected the results. 

Response 100: Thanks for pointing it out. We added relevant contents in the discussion section. “As a coincidence, the groups with the robots started with more floor eggs than the control group. Laying behavior of hens is extensively influenced by that of other birds in the same pen, and floor laying by the first few birds to come into lay could lead to high levels of floor eggs [31]. Although the proportional decrease of higher floor eggs in the robot groups was measured, the higher floor eggs and subsequent additional floor egg inducing may downgrade the robot effects on floor egg reduction. That could be one of the reasons for insignificant floor egg rate differences among the robot treatment group and control group.” (L471-L477)

Finally, it is not clear why the study has not been carried out from the begin of lay onwards, where floor eggs usually are a problem. So it also calls for further research. 

Response 101: We did want to test younger birds at first. However, farm managers recommended us to transport older birds since they get used to new environment faster and maintain better production performance. Younger birds could be sensitive to environment changes during transportation, and inconsistent environments among the original farm, transportation truck, and experimental house may cause physiological stress for birds and influence subsequent production performance. (L484-L487) We wanted to balance various aspects in this comprehensive project, and younger birds can be further researched in the future.

There are many tables and figures included, of which many provide similar information. The paper would benefit from reducing tables and figures, presenting the most important information, and removing the subheadings in the discussion section.

Response 102: Thank you for the suggestions. We have removed Figs. 3, 4, 5, and 8 and described relevant contents in the main text to avoid providing similar information. Meanwhile, we would kindly request for maintaining the subheadings in the discussion section, because the subheadings can separate the discussion with various topics and make the structure of discussion section clear.

Some specific comments:

Line 52-53, restriction of litter access may have negative welfare consequences, please add

Response 103: The sentence was changed to “Reasonable restriction of litter access has also been tried [4] but may have negative welfare consequences.” (L58-L59)

Line 60, I think these references are not in the proper format

Response 104: The format was changed to “Yang et al. [10]…” and “Parajuli et al. [11]…” (L66-L68)

Line 75 as far as I understand, earlier studies were also based on broiler chickens. So therefore it might also be interesting to include laying hens, as has been done here

Response 105: Relevant contents were changed to “…they were primarily based on broilers and observation of behavior, such as bird avoidance distance to robot [11]. Physiological stress indicators (i.e., serum corticosterone concentration [13]) were measured in this study to examine the potential robot-induced stress for laying hens.” (L86-L89)

Line 82 should be ‘reduced floor eggs more’ I suppose

Response 106: It was changed to “…reduced floor eggs more” (L96)

Line 83 I suppose nest box restriction was done to promote the occurrence of floor eggs, so that the effect could be measured.

Response 107: The content was added, “the nest box restriction may also promote the occurrence of floor eggs, so that the effect could be measured.” (L98-L99)

It is nice to have many pictures, but one of the pen and one of the robot is sufficient

Response 108: Following this suggestion, we combined these pictures into the fig. 1 containing one of the pen and one of the robot.

Resutls: please reduce number of figures and tables, especially where these present similar information. If there is no effect or difference, text might also be sufficient for some indicators, such as for foot pad health (all scores 0, no table needed). Eg. Mortality can be included in the text.

Response 109: We removed the previous figs. 4, 5, and 8 in the result section to avoid presenting similar information. However, we also kindly requested for maintaining the table results for footpad score and mortality, because the poultry science statistician in our research team suggested that the table presentation is necessary and straightforward for those results even though they are not significantly different. We hope the reviewer could understand.

Fig 11 remove from discussion section. If there is a figure it belongs to the results.

Response 110: The fig 7 (fig 11 previously) is a necessary support and evidence for anecdotal observation of bird adaptation to the robot. Meanwhile, we did not have sufficient data to conduct statistical analysis for bird adaptation to the robot, therefore, the figure is not statistically sound to be placed in the result section. As a result, we would like to kindly request for leaving this figure in the discussion section. 

Line 394 and further, yes of course, resources can be limited but within the period of study, the robots could also have been present continuously. So why was this not done?

Response 111: The whole experiment including the robot running procedure were planned and scheduled before it was started. We need to strictly execute the experimental plan to obtain statistical results, even though we observed that more robot running may reduce floor eggs more. Therefore, we call for further experiment to verify this hypothesis. (L448-L451)

Line 404 and further can be shortened a lot not really relevant to the paper

Response 112: We removed the contents per suggestion.

General production performance for Hy-Line Brown hens in weeks 34-43 is 90-95% for hen-day egg production, 0.9-1.6% for mortality, 1.85-2.00 kg/bird for live body weight, 108-114 g/bird/day for feed intake, 60.5-63.9 g/egg for egg mass, and 1.42-1.46 kg feed/dozen eggs for FCR.

Line 439 previous studies were on broilers, these on layers, I understood from your introduction

Response 113: One of the studies from Parajuli et al. [11] was also conducted for laying hens. (L68)

Line 448 and further, if bones are already mature, why then include this measure? Why not start at an earlier age with the trial?

Response 114: 

1. We expected robot can encourage more bird activity and further improve hen bone quality. We don’t know whether the hen bone quality can be improved before this experiment. The relevant content is an interpretation of the bone quality result rather than a prior knowledge with robot effects.

2. We did want to test younger birds at first. However, farm managers recommended us to transport older birds since they get used to new environment faster and maintain better production performance. Younger birds could be sensitive to environment changes during transportation, and inconsistent environments among the original farm, transportation truck, and experimental house may cause physiological stress for birds and influence subsequent production performance. (L486-L487) We wanted to balance various aspects in this comprehensive project, and younger birds can be further researched in the future.

3. We should point out that conducting such a comprehensive project is not easy, and we need to balance many aspects. We did want to measure young cage-free birds at first. However, Mississippi State does not have cage-free housing farms that can meet our requirement, and we need to seek the collaboration from the nearest state, Kentucky State. We need to drive around 6 hours to pick up birds, and based on the biosecurity protocol in the farm, we also need to leave the farm by 4:00 am, at which the weather was cold during winter. Considering cold stress, transportation stress, and adaptation to new housing, the experienced farm managers suggested us to use older birds that may more likely to overcome these challenges and recover faster after transportation than younger birds.

---

## [Decision Letter · Decision Letter 1]

23 Dec 2021

PONE-D-21-20853R1Effects of ground robots on hen floor egg reduction, production performance, stress response, bone quality, and behaviorPLOS ONE

Dear Dr. Guoming Li,

Thank you for submitting your manuscript to PLOS ONE. After careful consideration, we feel that it has merit but does not fully meet PLOS ONE’s publication criteria as it currently stands. Therefore, we invite you to submit a revised version of the manuscript that addresses the points raised during the review process.

We look forward to receiving your revised manuscript.

Kind regards,

Balamuralikrishnan Balasubramanian

Academic Editor

PLOS ONE

Additional Editor Comments:

Authors should pay more attention to revise based the reviewer's comments.

Highlight the revised sections.

Suggested to revise the manuscript once again for grammatical errors and to improve language efficiency.

The manuscript should be proofread by native speaker for the correction of English language so as to meet the standard of publication in the journal.

Reviewers' comments:

Reviewer's Responses to Questions

**Comments to the Author**

1. If the authors have adequately addressed your comments raised in a previous round of review and you feel that this manuscript is now acceptable for publication, you may indicate that here to bypass the “Comments to the Author” section, enter your conflict of interest statement in the “Confidential to Editor” section, and submit your "Accept" recommendation.

Reviewer #1: (No Response)

Reviewer #2: (No Response)

2. Is the manuscript technically sound, and do the data support the conclusions?

Reviewer #1: Yes

Reviewer #2: Yes

3. Has the statistical analysis been performed appropriately and rigorously? 

Reviewer #1: Yes

Reviewer #2: Yes

4. Have the authors made all data underlying the findings in their manuscript fully available?

Reviewer #1: Yes

Reviewer #2: Yes

5. Is the manuscript presented in an intelligible fashion and written in standard English?

Reviewer #1: No

Reviewer #2: No

6. Review Comments to the Author

Reviewer #1: The authors have made many revisions to the manuscript to address previous comments, however I still have a few queries/edits.

The quality of the scientific writing is not suitable for publication and I encourage the (presumably) native English co-authors to review the added text.

Lines 41-43: The first part of the sentence you make a general statement, the second part is specific to the experiment, please check and revise.

Line 54: on the littered floor.

Line 78: may have potential for reducing

Line 83: on commercial farms.

Lines 83-84: I don’t think you have to state what the benefits of a controlled experiment are. Most researchers will know the differences between a controlled trial and one on a commercial farm.

I’m not sure stating the measures were borrowed from broilers is correct. You can still measure these in laying hens, they are part of laying hen welfare assessments. My previous issue was why only these measures when you are testing laying hens not broilers. Also this new section needs to be checked to make sure the grammar is correct.

Lines 80-105. I’m not sure about the layout of this section. It is strange to see such a lengthy description of essentially your methods. Could this section be split up and incorporated into the above text to have a description of previous findings followed by what is missing. Then you can finalise the section with a statement of your objectives. Also, the writing could benefit from a proper check by the co-authors who I presume are native English speakers. The writing is not as good as it should be for publication.

Line 112: one commercial farm or multiple commercial farms. Could you include some explanation here as to why you used birds of 34 weeks and not young pullets. While I appreciate your response to my previous comment, other readers need to see some explanation of why such old birds were used.

Line 114: from a commercial farm

Line 120: enriched not enrich

Line 126-130: Same comment as previously, the new text needs to be read by the native speakers as the grammar is not correct. Likely this comment applies throughout the manuscript to all new text that has been added.

Line 142: each not wach. I recommend a more detailed check of your manuscript before resubmitting it again.

Line 151: ‘explained later’ reads colloquially. Please refer to a specific section and reword.

Line 165: ‘footpad’ rather than just ‘foot’ if you observed the footpad only.

Line 167: why only in flock 2?

Line 168: were weighed weekly.

Lines 177-183: an explanation for including this measurement is better placed within the introduction, although not with this exact wording in the intro.

Line 184: I’m not completely convinced by your explanation of ‘too many measurements’. If you are assessing a bird for their footpads, you can easily do a complete WQ assessment on that same bird in approximately 1 min (I’ve done plenty myself). I’ve run many laying hen trials that encompass many measurements. Perhaps the wording could be changed here to state you focussed on measures you predicted to be most affected by your treatment. Please avoid colloquial text, please get all additional text checked by a native speaker as the scientific writing is currently not of a standard for publication.

Lines 195-196: you need a clearer explanation of what was being observed in terms of ‘bird health status’ and why. Clarify the observers were assessing potential stress levels of the birds.

Lines 218-221: please check grammar, the wording seems strange to me. Unsure if there is a part missing in this sentence. ‘and then the remaining in a crucible’ ?

Line 320: how was it validated? Do you have data on that? Are you sure there was no detection range for your passive tag? Are you claiming the passive tags could be read at any distance? If they were validated to only be recording hens in the nest box then I presume there was actually a detection range for them.

Line 230: please state ‘attached’ rather than ‘bonded’. Bonded implies the tag was stuck to the leg, not attached to the leg.

Line 260: were used to reflect bird stress responses to the robot

Lines 291-195: Are these total sample sizes you are listing or per pen? Please make sure it is clear for the reader.

Line 297: So then did you remove non-significant interactions from your final models?

Line 325: footpad

Line 473: is nest use really a side effect in this trial? Also, the use of ‘side effect’ is colloquial.

Discussion: I did not intend for you to take my previous comments word for word and add them into your manuscript. Please check through the discussion for colloquial wording to improve the scientific writing quality.

Reviewer #2: First of all I would like to point out that the authors did a great job to try to answer and include the comments of the reviewers in the revised version of the paper. However, this not resulted in a better version of the paper which I will try to explain below. I am afraid the current version has not the quality needed of a version which can be published. I would advise the authors to let the paper be screened by experienced writers (senior researchers) both for language and scientific content (esp. the discussion section, which is long, and not to the point at many places). I give some examples below.

Line 30: adding that it now has 4 replicates still does not clarify the setup. I would like to see here a better explanation: each flock has 6 pens with 3 treatments, resulting in 2 replicates per treatment per flock. As a total of 2 flocks was studied, each treatment had a total of 4 replicates. Please adjust the text.

Line 12, same here. Please indicate that this were two successive flocks, and that in each flock 6 pens were present for four treatments. so then it is clear how you end up with 4 replicates.

Line 28 ‘human’ should be ‘humans’ . I have the feeling that the text has not been checked by a native speaker and I would encourage the authors to do so. There are more textual errors, I might have missed errors but please check the text.

Line 32, typo, chores should be scores

Line 42 same, wach should be each

Line 75 mutually verified. It is not clear to me what you mean herel.

Lie 84 I am happy with a detailed answer to the question, but this argument should not have a place here but in the introduction and the text should be shortened. Further I am not convinced by this answer and simply stating that you had too many measures and should select one is not a good argument. You can better argue why you think this specific parameter is affected by the robot.

Line 262 replace to build by ‘Week 34 corticosterone concentrations were considered as baseline concentrations after adaptation to the new housing’

Statistical paragraph: sample size can be included in the results. This should be described better, you can simply say that log transformation was used to approach normal distribution.

Line 478 remove ‘and’ at the beginning of the sentence.

Line 482 should be floor egg percentage and indicate should be indicates

Line 301 and further, this should not be in the discussion. I understand your experiment was planned, but this argumentation is not convincing.

Line 315. I think there are behaviors that can indicate whether or not hens are used to the robot. Distance to robot is one, also home pen behaviour (activity, restlessness) and fear responses can be used. Please remove this text and focus on the avoidance behavior (which is an appropriate measure)

Line 405 this should not be part of the discussion. I expect here a comparison with other findings and if these are not present, a critical reflection on the result

The discussion has not been improved as compared to the first version of the paper, unfortunately. Please keep it concise, use appropriate literature if present, and try to limit it to placing your findings in the context of what already has been found. It is now difficult to read, contains information which should not be there.

Please remove the figure from the discussion. Place it in the results or in the supporting information (better option) and briefly mention the data.

7. PLOS authors have the option to publish the peer review history of their article (what does this mean?). If published, this will include your full peer review and any attached files.

Reviewer #1: No

Reviewer #2: No

---

## [Author Response · Author response to Decision Letter 1]

31 Jan 2022

Dear editor and reviewers,

On behalf of the co-authors, we are very grateful for the valuable comments from you and for giving us the

opportunity to further improve our manuscript. To be honest, we were overwhelmed by the 114 comments in the previous review and focused most of our attention on addressing the comments technically. We also realized our language may not meet the requirement of publications; therefore, we improved the relevant contents proposed by the reviewers and invited the native English speakers within the research group to proofread the paper. The revised portions can be found in the track-change manuscript, and the line numbers in this response file are referred to in the clean-version manuscript. 

Special thanks to the Reviewers for their valuable remarks and comments again. We also earnestly appreciate the Editor’s diligent work. We hope that the corrections will meet your approval and our paper

will be finally accepted for publication in PLOS ONE. Looking forward to hearing from you!

Best Regards,

Guoming Li.

Redundancy and readability. 

Additional Editor Comments

Authors should pay more attention to revise based the reviewer's comments. 

Response 1: We have carefully gone through the reviewers’ comments again and made sure that everything is correct and accurate. 

Highlight the revised sections.

Response 2: The previous revisions were highlighted in yellow text, while the current revisions were highlighted in blue font.

Suggested to revise the manuscript once again for grammatical errors and to improve language efficiency. The manuscript should be proofread by native speaker for the correction of English language so as to meet the standard of publication in the journal.

Response 3: Grammatical errors and language efficiency were double-checked and proofread by the native English speaker within the research group.

Reviewer #1: 

The quality of the scientific writing is not suitable for publication and I encourage the (presumably) native English co-authors to review the added text.

Response 4: We have invited the native English co-authors to review the added text.

Lines 41-43: The first part of the sentence you make a general statement, the second part is specific to the experiment, please check and revise.

Response 5: The second part was removed to avoid any confusion.

however, such an effect diminished as hens became familiarized with the robot

Line 54: on the littered floor.

Response 6: It was changed to ‘…on the littered floor’ (L48)

Line 78: may have potential for reducing

Response 7: Relevant contents were removed to improve language accuracy. 

and robotics may have potential for reducing floor eggs of hens in CF housing systems but should be evaluated

Line 83: on commercial farms.

Response 8: It was changed to ‘…on commercial farms’ (L76).

Lines 83-84: I don’t think you have to state what the benefits of a controlled experiment are. Most researchers will know the differences between a controlled trial and one on a commercial farm.

Response 9: The statement of describing benefits of floor pens was removed.

Therefore, floor pens were set for the preliminary examinations because of controlled environments, convenient management with limited labor, and so on.

I’m not sure stating the measures were borrowed from broilers is correct. You can still measure these in laying hens, they are part of laying hen welfare assessments. My previous issue was why only these measures when you are testing laying hens not broilers. Also this new section needs to be checked to make sure the grammar is correct.

Response 10: Thanks for pointing this out. We rephrased the contents as follows. And grammar was double-checked to ensure language accuracy. 

“Robot manipulation may encourage hen walking behaviors and enhance litter aeration via me-chanical disturbance of litter shavings during turning operations as a result of skid steer locomotion and the associated traction force dynamics experienced while turning. Litter moisture may be re-duced due to decreased hen lying time and improved ventilation in the aerated litter. While not commonly measured in a short-term trial for laying hens, litter moisture content differed signifi-cantly from various treatments for 34-week laying hens in an eight-week experiment [17]. Hen footpad health may benefit from the potentially improved litter conditions and can be reflected by one of the representative footpad measures [18] like footpad score determined by footpad derma-titis rates [19]. Although young hens are not likely to have footpad dermatitis, Campbell et al. [20] reported a 0.3% footpad dermatitis rate for free-range hens at 20-36 weeks of age.” (L90-L100)

Lines 80-105. I’m not sure about the layout of this section. It is strange to see such a lengthy description of essentially your methods. Could this section be split up and incorporated into the above text to have a description of previous findings followed by what is missing. Then you can finalise the section with a statement of your objectives. Also, the writing could benefit from a proper check by the co-authors who I presume are native English speakers. The writing is not as good as it should be for publication.

Response 11: We have revised the section according to this comment. The section was split up into one paragraph describing the necessity of the measures and another paragraph to state the objective of this study. Grammar was checked and details can be found in L75-L105.

Line 112: one commercial farm or multiple commercial farms. 

Response 12: The birds were from two commercial farms. (L111)

Could you include some explanation here as to why you used birds of 34 weeks and not young pullets. While I appreciate your response to my previous comment, other readers need to see some explanation of why such old birds were used.

Response 13: Some descriptions were added to indicate the reason of selecting 34-week-old birds rather than young pullets.

“Older birds (34 weeks of age) rather than young pullets were recommended by the industry farm managers, because they may be more likely to overcome the challenges of transportation stress, cold stress (at the end of October), and adaptation to new housing environments.” (L113-L116)

Line 114: from a commercial farm

Response 14: It was changed to “..from two commercial farms”. (L111)

Line 120: enriched not enrich

Response 15: It was changed to ‘enriched’. (L121)

Line 126-130: Same comment as previously, the new text needs to be read by the native speakers as the grammar is not correct. Likely this comment applies throughout the manuscript to all new text that has been added.

Response 16: All added texts were proofread by the native speakers. Changes can be found in the track-change manuscript.

Line 142: each not wach. I recommend a more detailed check of your manuscript before resubmitting it again.

Response 17: It was changed to ‘each’. (L141)

Line 151: ‘explained later’ reads colloquially. Please refer to a specific section and reword.

Response 18: It was changed to ‘Section 2.3.4 Nesting behavior monitoring’ (L149-L150)

Line 165: ‘footpad’ rather than just ‘foot’ if you observed the footpad only.

Response 19: It was changed to ‘footpad’ throughout the manuscript.

Line 167: why only in flock 2?

Response 20: The data collection was conducted in the two flocks. However, due to unexpected events, egg mass data in flock 1 was lost, and only flock 2 data were used for further analysis. (L163-164)

Line 168: were weighed weekly.

Response 21: It was changed to ‘…were weighed weekly’ (L165).

Lines 177-183: an explanation for including this measurement is better placed within the introduction, although not with this exact wording in the intro.

Response 22: The explanation for including the litter moisture content measurement was placed within the introduction and rephrased. Details can be found in L75-L100. 

Line 184: I’m not completely convinced by your explanation of ‘too many measurements’. If you are assessing a bird for their footpads, you can easily do a complete WQ assessment on that same bird in approximately 1 min (I’ve done plenty myself). I’ve run many laying hen trials that encompass many measurements. Perhaps the wording could be changed here to state you focussed on measures you predicted to be most affected by your treatment. Please avoid colloquial text, please get all additional text checked by a native speaker as the scientific writing is currently not of a standard for publication.

Response 23: We may mislead the reviewer about stating ‘too many measurements’. Yes, measuring footpad of one hen was really quick, however, besides that, we need to euthanize birds, take bone samples, obtain the weights of feed, eggs, and birds, and count eggs at the end of this experiment. The farm manager only allowed us to visit the farm and complete these tasks within the same day because of biosecurity issue. That means a lot for us. Therefore, we selected one of the representative measures, footpad score, to reflect footpad health while avoiding overloading. 

Similar to litter moisture content, we also moved the description about rationale of selecting the footpad score measure to introduction and rephrased the contents. Colloquial texts were rephrased to meet the standard of publication. Detailed rationale of the two measures can be found in the introduction section. can be found in L90-L100.

Lines 195-196: you need a clearer explanation of what was being observed in terms of ‘bird health status’ and why. Clarify the observers were assessing potential stress levels of the birds.

Response 24: The poultry scientist in our research group helped us to clarify that. 

“All birds for blood sampling were handled similarly to avoid introducing additional stress during collection. The physiological parameters (i.e., heart rate, respiratory rate, body temperature, and mucous membrane color) were observed and detected by the poultry scientists before blood sam-pling. This approach ensured that all assessed birds were healthy and minimized confounding factors or pathological conditions that would otherwise affect the results.” (L177-L182)

Lines 218-221: please check grammar, the wording seems strange to me. Unsure if there is a part missing in this sentence. ‘and then the remaining in a crucible’ ?

Response 25: The sentences were revised to the following (L201-203).

“The oven-dried bone was weighed, placed in a crucible, and oven-dried again in a furnace (Isotemp D3714 muffle furnace, Thermo Fisher Scientific Inc., Waltham, MA) for 24 hours at 600 °C. After the last drying, the remaining content was weighed to obtain ash weights of bones.”

Line 320: how was it validated? Do you have data on that? Are you sure there was no detection range for your passive tag? Are you claiming the passive tags could be read at any distance? If they were validated to only be recording hens in the nest box then I presume there was actually a detection range for them.

Response 26: After the hardware was installed, an RFID tag was held near the antenna, and d the interface of a free visualization software, MultiReader for SpeedWay Gen2 RFID (Version 6.6.11.240), was ob-served. Once the RFID tag was within the detection range of the antenna, readings were displayed on the interface and the system power was adjusted, so that the system was validated to only record birds located inside a nest box rather than birds standing next to a nest box. The free software was only for real-time data visualization and could not store validation data. It should be noted that passive tags neither have an internal power source nor actively emit signals. They utilize electro-magnetic waves received from a reader. Once the reader transmits signals to a tag, the connected antenna creates a magnetic field and the tag circuit uses the power generated to transmit data back to the reader. Theoretically, a passive tag does not have a detection range, but the connected antenna had one within 80 cm based on current RFID settings and power supply (L211-L221). Therefore, we just need to validate the power of the system to ensure the antenna atop a nest box for only covering hens in the box. 

Line 230: please state ‘attached’ rather than ‘bonded’. Bonded implies the tag was stuck to the leg, not attached to the leg.

Response 27: Thanks. It was changed to ‘attached’. (L210)

Line 260: were used to reflect bird stress responses to the robot

Response 28: It was changed to ‘…were used to reflect bird stress responses to the robot’. (L249).

Lines 291-195: Are these total sample sizes you are listing or per pen? Please make sure it is clear for the reader.

Response 29: It was changed to ‘The sample size (n) for each treatment per pen was…’ (L273).

Line 297: So then did you remove non-significant interactions from your final models?

Response 30: We did not remove non-significant interactions from the final models but just not report the interaction results. (L282)

Line 325: footpad

Response 31: It was changed to ‘footpad’ throughout the manuscript.

Line 473: is nest use really a side effect in this trial? Also, the use of ‘side effect’ is colloquial.

Response 32: The ‘side effect’ was removed from the article. The sentence was changed to “This research was mainly aimed to investigate the effects of ground robot manipulation on hen floor egg reduction, production performance, stress response, bone quality, and nesting behaviors.” (L444-445)

Discussion: I did not intend for you to take my previous comments word for word and add them into your manuscript. Please check through the discussion for colloquial wording to improve the scientific writing quality.

Response 33: We understand that the reviewer does not intend to ask us for including the comment words, but we thought these wordings can strengthen the scientific values of the discussion. Therefore, we would kindly request these wordings to be included into the discussion. But if this causes any ethical problem, we will remove the relevant contents. The whole discussion section was rephrased to enhance the article quality.

Reviewer #2: 

First of all I would like to point out that the authors did a great job to try to answer and include the comments of the reviewers in the revised version of the paper. However, this not resulted in a better version of the paper which I will try to explain below. I am afraid the current version has not the quality needed of a version which can be published. I would advise the authors to let the paper be screened by experienced writers (senior researchers) both for language and scientific content (esp. the discussion section, which is long, and not to the point at many places). I give some examples below.

Response 34: We appreciated the reviewer’s compliment on our previous revision and realized that we did not do well in the language. The senior researchers in the research group have proofread this article and made sure everything is accurate and concise. 

Line 30: adding that it now has 4 replicates still does not clarify the setup. I would like to see here a better explanation: each flock has 6 pens with 3 treatments, resulting in 2 replicates per treatment per flock. As a total of 2 flocks was studied, each treatment had a total of 4 replicates. Please adjust the text.

Response 35: Thanks. The relevant contents were rephrased according to the reviewer’s comment. 

“Two successive flocks of 180 Hy-Line Brown hens at 34 weeks of this age were used. The treatment structure for each flock consisted of six pens with three treatments (without robot running, with one-week robot running, and with two-weeks robot running), resulting in two replicates per treatment per flock and four replicates per treatment with two flocks.” (L26-L30)

Line 12, same here. Please indicate that this were two successive flocks, and that in each flock 6 pens were present for four treatments. so then it is clear how you end up with 4 replicates.

Response 36: The ‘two successive flocks’ and ‘six pens in each flock’ are added and indicated in L27-29.

Line 28 ‘human’ should be ‘humans’ . I have the feeling that the text has not been checked by a native speaker and I would encourage the authors to do so. There are more textual errors, I might have missed errors but please check the text.

Response 37: It was changed to ‘technicians’ (L128).

Line 32, typo, chores should be scores

Response 38: To avoid confusion, the ‘chores’ was changed to ‘tasks’ (L131)

Line 42 same, wach should be each

Response 39: ‘wach’ was changed to ‘each’. (L141)

Line 75 mutually verified. It is not clear to me what you mean herel.

Response 40: To avoid confusion, the sentence was changed ‘…evaluation results were verified between observers’ (L171-L172)

Lie 84 I am happy with a detailed answer to the question, but this argument should not have a place here but in the introduction and the text should be shortened. Further I am not convinced by this answer and simply stating that you had too many measures and should select one is not a good argument. You can better argue why you think this specific parameter is affected by the robot.

Response 41: We rephrased the rationale statements of selecting footpad scores and litter moisture content and place them in the introduction. (L90-L100)

Robot manipulation may encourage hen walking behaviors and enhance litter aeration via me-chanical disturbance of litter shavings during turning operations as a result of skid steer locomotion and the associated traction force dynamics experienced while turning. Litter moisture may be re-duced due to decreased hen lying time and improved ventilation in the aerated litter. While not commonly measured in a short-term trial for laying hens, litter moisture content differed signifi-cantly from various treatments for 34-week laying hens in an eight-week experiment [17]. Hen footpad health may benefit from the potentially improved litter conditions and can be reflected by one of the representative footpad measures [18] like footpad score determined by footpad derma-titis rates [19]. Although young hens are not likely to have footpad dermatitis, Campbell et al. [20] reported a 0.3% footpad dermatitis rate for free-range hens at 20-36 weeks of age.

Line 262 replace to build by ‘Week 34 corticosterone concentrations were considered as baseline concentrations after adaptation to the new housing’

Response 42: The sentence was changed to “Week 34 corticosterone concentrations were considered baselines after birds adapted to the new housing”. (L174-L175)

Statistical paragraph: sample size can be included in the results. This should be described better, you can simply say that log transformation was used to approach normal distribution.

Response 43: The sample size were indeed included in each table notes. And the other reviewer required us to summarize the sample size, and we may kindly request to maintain these sample size summary in the statistical paragraph. 

The relevant contents were revised to “… approach normal distribution”. (L272)

Line 478 remove ‘and’ at the beginning of the sentence.

Response 44: ‘And’ was removed. (L450).

And placing them into new…

Line 482 should be floor egg percentage and indicate should be indicates

Response 45: The sentence was changed to ‘…high floor egg rates…’ (L451).

Line 301 and further, this should not be in the discussion. I understand your experiment was planned, but this argumentation is not convincing.

Response 46: We understand the reviewer’s concern about more robot manipulation resulting in less floor eggs. However, like we debate previously that ‘The whole experiment including the robot running procedure were planned and scheduled before it was started. We need to strictly execute the experimental plan to obtain statistical results, even though we guessed that more robot running may reduce floor eggs more’. Although this argumentation is not convincing, the only way we can do is to call for further experiment to verify this hypothesis. To avoid colloquial texts, we also removed relevant contents.

Line 315. I think there are behaviors that can indicate whether or not hens are used to the robot. Distance to robot is one, also home pen behaviour (activity, restlessness) and fear responses can be used. Please remove this text and focus on the avoidance behavior (which is an appropriate measure)

Response 47: We added some discussion, “Other behaviors, such as activity and restlessness, and fear responses may also be examined in the future for indicating bird adaptations to robots. (L477-478)

Line 405 this should not be part of the discussion. I expect here a comparison with other findings and if these are not present, a critical reflection on the result

Response 48: We are sorry that we cannot be navigated to the exact problem.

Around the Line 405 of the previous clean-version manuscript, “It should be noted that the frequency was only counted when there was at least one bird using a nest box, and the frequency for the empty nest box was not included.” are presented in the result section.

Around the Line 405 of the previous track-change manuscript, “Table 6 shows the means of parameters for evaluating bone quality of hens. The average bone breaking force, fresh bone weight, dried bone weight, bone ash weight, and ash percentage were 24.8 kg, 11.0 g, 7.3 g, 4.1 g, and 56.0%, respectively, and they were not significantly different with the treatments (P≥0.20).” are presented in the result section.

Neither are these two places part of the discussion. If the reviewer can pinpoint this question, we would love to revise it accordingly.

The discussion has not been improved as compared to the first version of the paper, unfortunately. Please keep it concise, use appropriate literature if present, and try to limit it to placing your findings in the context of what already has been found. It is now difficult to read, contains information which should not be there.

Please remove the figure from the discussion. Place it in the results or in the supporting information (better option) and briefly mention the data.

Response 49: We reorganize the discussion section and proofread it. The figure indicating the bird adaptation and relevant descriptions were moved to the result sections. Although the reviewer suggested placing the figure in the supporting information be a better option, we may think the figure is informative about the adaptation process. Therefore, we kindly request to maintain this in the paper (result section). Details can be found in the result section, “Anecdotal observations for bird adaptation to the robot”.

---

## [Decision Letter · Decision Letter 2]

22 Mar 2022

PONE-D-21-20853R2Effects of ground robot manipulation on hen floor egg reduction, production performance, stress response, bone quality, and behaviorPLOS ONE

Dear Dr. Guoming Li

Thank you for submitting your manuscript to PLOS ONE. After careful consideration, we feel that it has merit but does not fully meet PLOS ONE’s publication criteria as it currently stands. Therefore, we invite you to submit a revised version of the manuscript that addresses the points raised during the review process.

We look forward to receiving your revised manuscript.

Kind regards,

Balamuralikrishnan Balasubramanian

Academic Editor

PLOS ONE

Journal Requirements:

Reviewers' comments:

Reviewer's Responses to Questions

**Comments to the Author**

1. If the authors have adequately addressed your comments raised in a previous round of review and you feel that this manuscript is now acceptable for publication, you may indicate that here to bypass the “Comments to the Author” section, enter your conflict of interest statement in the “Confidential to Editor” section, and submit your "Accept" recommendation.

Reviewer #1: (No Response)

Reviewer #2: All comments have been addressed

2. Is the manuscript technically sound, and do the data support the conclusions?

Reviewer #1: Yes

Reviewer #2: Yes

3. Has the statistical analysis been performed appropriately and rigorously? 

Reviewer #1: Yes

Reviewer #2: Yes

4. Have the authors made all data underlying the findings in their manuscript fully available?

Reviewer #1: Yes

Reviewer #2: Yes

5. Is the manuscript presented in an intelligible fashion and written in standard English?

Reviewer #1: Yes

Reviewer #2: Yes

6. Review Comments to the Author

Reviewer #1: Thank you for addressing the previous comments. I have a few further minor edits on the text.

Line 51: If not rapidly collected OR If not collected in a timely manner

Line 63: I presume this should be ‘has not yet’

Line 64: Researchers have investigated (this would read better)

Line 77: preliminary trials were conducted in experimental pens

Line 78: ‘performance of floor eggs’ is strange wording. Could it be adjusted to be ‘performance of floor egg laying’ or ‘interested in floor egg laying and other production indicators’

Lines 81-85: please reword this sentence, it is long and awkwardly worded

Lines 75-100: I think this whole section could be streamlined. It is currently very wordy and the points could be made more succinctly. Litter effects may make more sense placed after the bone quality statement as both mention hen movement.

Line 85: It is not clear whether you mean the robot operation itself is a visual/audio alert, or if you are stating this is something that the robot does – i.e., it emits specific sounds to trigger the hens to use the laying boxes?

Line 102: specifically nesting behavior only? You mention hen activity in the previous section, but you didn’t directly measure that behavior (from memory?), so clarify it is just nesting behavior here in the objectives.

Line 127: ‘adverse’ would be a better word than ‘inappropriate’

Line 195: At the end of the experiments,

Line 196: ‘after euthanasia’ is not necessary, you already state they were euthanized,

Line 212: is there a stray ‘d’ that needs to be removed?

Line 223: and stored in a Python-based

Lines 269-270: A constant of 1 was added to all percentage data in decimal form to eliminate negatives and then log transformed to approach normality. (you can then delete the sentence 271-272 about the log transformation

Line 282: are not reported

Line 287: had no effect

Line 349: Data were

Line 401: The overall trend showed the time in

Line 426: The overall trend with the three treatments showed hourly

Line 437: At the end of the two-week robot running period,

Line 444: ‘mainly aimed’? Or ‘aimed’? You have listed all your measurements there so not sure why it is stated as ‘mainly aimed’

Line 449: I would think it is ‘likely’ rather than ‘may’ have established laying patterns

Line 527: up to two weeks after robot running.

Line 552: would it be better to state ‘less than a previous study using the same RFID system’? If that is what you are referring to. When you state ‘the previous study’, it implies the reader should know the specific study you are referring to.

Reviewer #2: (No Response)

7. PLOS authors have the option to publish the peer review history of their article (what does this mean?). If published, this will include your full peer review and any attached files.

Reviewer #1: No

Reviewer #2: No

---

## [Author Response · Author response to Decision Letter 2]

24 Mar 2022

Dear editor and reviewers,

On behalf of the co-authors, I would like to thank you for your hard work again, especially for reviewer #1. I realized how much time the reviewer input for reviewing this manuscript and appreciated his/her insightful comments helping us improve language accuracy in this round. I enjoyed working on these comments and interacting with the reviewer and learned a lot from them. The changes are highlighted in the manuscript, and the response are in blue font in this file.

Best Regards,

Guoming Li.

Reviewer #1: 

Line 51: If not rapidly collected OR If not collected in a timely manner

Response 1: It was changed to ‘If not collected in a timely manner’ (L51).

Line 63: I presume this should be ‘has not yet’

Response 2: It was changed to ‘has not yet’ (L63). 

Line 64: Researchers have investigated (this would read better)

Response 3: It was changed to ‘Researchers have investigated’ (L64). 

Line 77: preliminary trials were conducted in experimental pens

Response 4: It was changed to ‘preliminary trials were conducted in experimental pens’ (L77).

Line 78: ‘performance of floor eggs’ is strange wording. Could it be adjusted to be ‘performance of floor egg laying’ or ‘interested in floor egg laying and other production indicators’

Response 5: It was changed to ‘performance of floor egg laying’ (L78-79).

Lines 81-85: please reword this sentence, it is long and awkwardly worded

Response 6: The sentence was revised as follows.

‘Yang et al. [10] elaborated that robot operations encouraged broiler movement, and increasing bird activities have been correlated to improved bird bone quality in loose housing systems [3, 14]. Therefore, the robot was assumed to increase hen activities in this study and consequently enhance bird bone quality.’ (L81-85).

Lines 75-100: I think this whole section could be streamlined. It is currently very wordy and the points could be made more succinctly. Litter effects may make more sense placed after the bone quality statement as both mention hen movement.

Response 7: We reorganized this paragraph and removed the redundancy of description. The paragraph is inevitably lengthy because we want to justify every measure is reasonable. The litter moisture contents were placed after the bone quality statement per suggestions. The reorganized paragraph can be found in L75-96.

Line 85: It is not clear whether you mean the robot operation itself is a visual/audio alert, or if you are stating this is something that the robot does – i.e., it emits specific sounds to trigger the hens to use the laying boxes?

Response 8: It was changed to ‘Ground robots can emit sounds and lights having visual and/or audible alert capabilities and encouraging birds to utilize nest boxes’ (L91-92).

Line 102: specifically nesting behavior only? You mention hen activity in the previous section, but you didn’t directly measure that behavior (from memory?), so clarify it is just nesting behavior here in the objectives.

Response 9: It was changed to ‘and nesting behavior’ (L98).

Line 127: ‘adverse’ would be a better word than ‘inappropriate’

Response 10: It was changed to ‘adverse’ (L122).

Line 195: At the end of the experiments,

Response 11: It was changed to ‘At the end of the experiments’ (L190). 

Line 196: ‘after euthanasia’ is not necessary, you already state they were euthanized,

Response 12: The words were removed. after euthanasia. (L191).

Line 212: is there a stray ‘d’ that needs to be removed?

Response 13: The ‘d’ was removed. 

Line 223: and stored in a Python-based

Response 14: It was changed to ‘and stored in a Python-based’ (218).

Lines 269-270: A constant of 1 was added to all percentage data in decimal form to eliminate negatives and then log transformed to approach normality. (you can then delete the sentence 271-272 about the log transformation

Response 15: It was changed to ‘A constant of 1 was added to all percentage data in decimal form to eliminate negatives and then log transformed to approach normality.’ Additional log transformation statement was removed. The log transformation was used to make the examined data approach a normal distribution. (L267-269) 

Line 282: are not reported

Response 16: It was changed to ‘are not reported’ (L277).

Line 287: had no effect

Response 17: It was changed to ‘had no effect’ (L285). 

Line 349: Data were

Response 18: It was changed to ‘Data were’ (L352).

Line 401: The overall trend showed the time in

Response 19: It was changed to ‘The overall trend with the three treatments showed that hourly time spent’ (L404-405).

Line 426: The overall trend with the three treatments showed hourly

Response 20: It was changed to ‘The overall trend with the three treatments showed that hourly’ (L429-430).

Line 437: At the end of the two-week robot running period,

Response 21: It was changed to ‘At the end of the two-week robot running period’ (L440).

Line 444: ‘mainly aimed’? Or ‘aimed’? You have listed all your measurements there so not sure why it is stated as ‘mainly aimed’

Response 22: That is correct, thanks. The ‘mainly’ was removed (L447). 

Line 449: I would think it is ‘likely’ rather than ‘may’ have established laying patterns

Response 23: It was changed to ‘likely to have already’ (L452). 

Line 527: up to two weeks after robot running.

Response 24: It was changed to ‘up to two weeks after robot running’ (L530).

Line 552: would it be better to state ‘less than a previous study using the same RFID system’? If that is what you are referring to. When you state ‘the previous study’, it implies the reader should know the specific study you are referring to.

Response 25: It was changed to ‘less than a previous study using the same RFID system’ (L555-556)

---

## [Editor Report · Decision Letter 3]

12 Apr 2022

Effects of ground robot manipulation on hen floor egg reduction, production performance, stress response, bone quality, and behavior

PONE-D-21-20853R3

Dear Dr. Guoming Li

We’re pleased to inform you that your manuscript has been judged scientifically suitable for publication and will be formally accepted for publication once it meets all outstanding technical requirements.

Kind regards,

Balamuralikrishnan Balasubramanian

Academic Editor

PLOS ONE